# Web-Based Serious Games and Accessibility: A Systematic Literature Review

**Luis Salvador-Ullauri [1], Patricia Acosta-Vargas [2],\* and Sergio Luján-Mora [1]** 

1   Department of Software and Computing Systems, University of Alicante, 03690 Alicante, Spain; lasu1@alu.ua.es (L.S.-U.); sergio.lujan@ua.es (S.L.-M.)
2   Intelligent and Interactive Systems Laboratory, Universidad de Las Américas, Quito 170125, Ecuador
\*   Correspondence: patricia.acosta@udla.edu.ec

**Abstract:** Nowadays, serious games, called training or learning games, have been incorporated into teaching and learning processes. Due to the increase of their use, the need to guarantee their accessibility arises in order to include people with disabilities in the educational environments in an integral way. There are reviews of the literature on video games but not on web-based serious games. Serious games are different from the previous ones because their educational processes allow reinforcing learning. This literature review was conducted using the recommendations for systematic reviews proposed by Kitchenham and Petersen. Three independent reviewers searched the ACM Digital Library, IEEE Xplore, Scopus, and Web of Science databases for the most relevant articles published between 2000 and 2020. Review selection and extraction were made using an interactive team approach. We applied the study selection process's flowchart adapted from the PRISMA statement to filter in three stages. This systematic literature review provides researchers and practitioners with the current state of web-based serious games and accessibility, considering cognitive, motor, and sensory disabilities.

**Keywords:** accessibility; games; literature; review; serious; systematic; Web Content Accessibility Guideline 2.1

## 1. Introduction

The Web has changed the way people communicate and relate to each other. Technology has generated a continuous impact on society and individuals' behavior. The increasing access to the Web and the variety of devices that allow us to interact with it have made it possible for students to choose the tools and services that best suit their needs, and thus to personalize the learning experience [1].

Figure 1 shows the Google Trends search related to web applications, serious games, and mobile applications made on the Web in the last five years. We found that the term serious games and web applications began to intensify from 2019.

Serious games are "games that do not have entertainment, enjoyment, or fun as their main objective" (p. 21, [3]). The main objectives of serious games can be, among others, education, training, human resources management, and health improvement [4]. Web-based serious games constitute an area growing thanks to the improvement of browsers and technologies used on the Web [1], which have reduced the gap between desktop and web applications.

According to Statista [5], the game-based learning market revenue worldwide between 2018 and 2024 indicates the serious games market is expected to grow from 3.5 billion U.S. dollars to 24 billion in 2024. The trend of serious web-based games has several benefits: (1) Reinforce learning in educational processes virtually and at a distance [1]. (2) Use the applications without the need to download, install, and configure. (3) Interact with the applications at any time and space. (4) Update the application

automatically with the latest version. (5) Use the applications with fewer technical problems due to software or hardware conflicts with other applications.

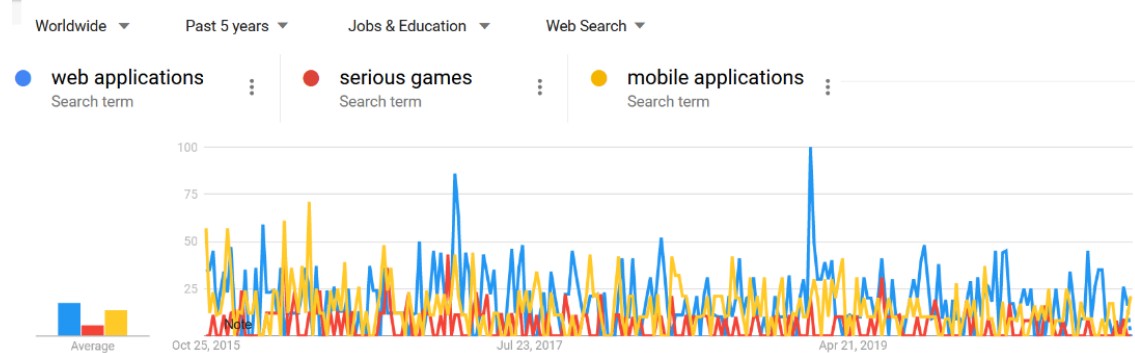

**Figure 1.** Trend of web-based serious games [2].

Nowadays, there are many web-based serious games, but developers are not usually worried about making them accessible. The accessibility [6] aims to ensure that applications can be used by the maximum number of people, regardless of their abilities and regardless of the technical characteristics of the equipment used to access the application. According to the Web Content Accessibility Guidelines (WCAG) 2.1, some guidelines help validate web pages, applications including serious games, making them more accessible to everyone, including people with disabilities.

This study compares articles related to accessibility in serious web-based games. In this research, we present a systematic literature review (SLR) [7,8] that allows examining serious games' accessibility. We start from the following research question: What accessibility evaluation standards have been used by developers to create serious web-based games? This study defines the query strings that allow finding the most significant research related to accessibility in serious games [9]. To determine the query string, we apply the structure in terms of population, intervention, comparison, and outcome (PICO) [10].

This SLR allowed us to (1) outline the issues relevant to serious web-based gaming and accessibility studies; (2) identify how accessibility is involved in serious gaming; (3) determine accessibility guidelines based on the WCAG, and (4) identify the assistive technologies and devices used to achieve accessibility in serious gaming according to disability. After an extraction process of 476 studies, a collection of 47 primary studies was selected using the Preferred Reporting Elements for Systematic Reviews (PRISMA) [11,12], the flowchart in the selection process.

This article is organized as follows. Section 2 introduces readers to the topic of accessibility and serious games. Section 3 describes the research method used for the systematic review of the literature. Section 4 includes the bibliometric analysis results and the literature review. Section 5 presents discussions of the results, along with limitations for research. Finally, Section 6 presents conclusions and future research work.

## 2. Background and Motivation

The formal description given by [13] indicates that "serious games" have an explicit and carefully thought out educational purpose and are not intended to be played primarily for fun [14]. Serious games are educational or training games [14], while video games [15] provide a cultural outlet where more players can be included and interacted to perform activities between different users.

This study identified video game SRL publications but not on serious games, so we justify this study's need to: (1) identify information on the most relevant research on web-based serious games and accessibility; (2) identify accessibility guidelines that apply to web-based serious games; (3) detect the different approaches to web-based serious games for cognitive, motor, and sensory disabilities; (4) identify the WCAG-based accessibility guidelines applied to serious games to determine trends and gaps in serious games development; (5) identify authors conducting accessibility studies on serious games.

*Serious Games and Accessibility*

Accessibility in serious games [16] aims to ensure that serious games can be used by the maximum number of people to access serious games. The authors suggest applying the four principles of WCAG 2.1.

Several studies [9,17] show a lack of commitment by designers to implement accessibility. For this reason, there is a low percentage of accessible serious games. Currently, serious games have been incorporated into the teaching-learning processes. Therefore, it is essential to guarantee accessibility [18] so that the largest number of people can use them.

The authors [19] present an analysis of the accessibility guidelines for the development of videogames; the study is oriented to cognitive disabilities. It also proposes categorizing the guidelines that should be used to analyze a video game's accessibility, especially serious games. The authors present an evaluation tool for the development of serious games for mobile devices.

Following the article, the authors [20] indicate that it is a great challenge to implement serious games to support the learning processes of people with cognitive disabilities. The authors evaluated ten serious video games using the design principles established by WCAG 2.0 [21]. The results revealed that applications do not reach an adequate level of accessibility to be used by people with cognitive problems. However, they do meet some of the accessibility requirements.

## 3. Method Applied for Systematic Literature Review

This SRL [7,8] began defining a review protocol, the research question, and the methods. In this SLR, we apply the PRISMA Statement, which consists of a list of 27 elements and a four-phase flow diagram. The PRISMA method is frequently used in health issues [22]; this method was adapted to identify studies related to accessibility and serious games. We attached a checklist (Appendix A); in PRISMA Checklist, we record the page number or pages in which compliance or non-compliance with the 27 items detailed in the seven sections can be evidenced: (1) Title, (2) Summary, (3) Introduction, (4) Methods, (5) Results, (6) Discussion, and (7) Funding.

Figure 2 shows the review process consisting of five phases: (1) definition of the research questions to review the scope; (2) search strategy to obtain all documents; (3) screening of the documents to extract the most relevant documents; (4) keywording using abstracts for the classification scheme, and (5) data extraction and revision process to obtain the results of the systematic review.

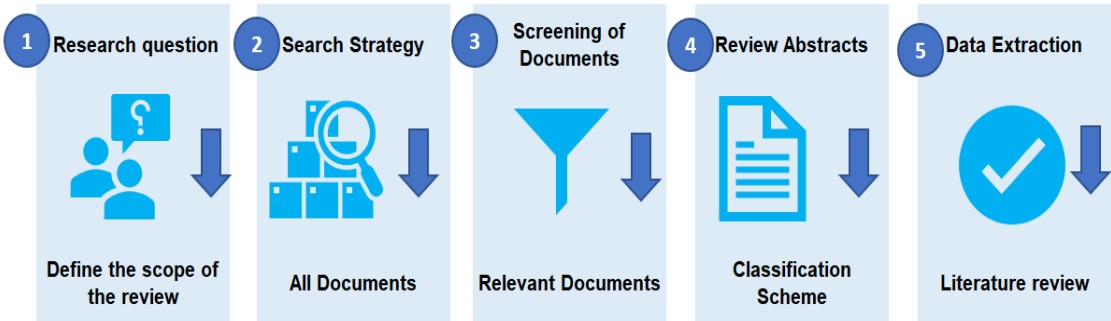

**Figure 2.** Review process.

### 3.1. Research Objectives and Questions

This study's first objective is to present information about the most relevant research on published web-based serious games and accessibility. This SLR contains a series of articles from the digital libraries and details the authors, the year of publication, and the Scimago Journal Rank (SJR) impact factor.

The second objective is to detect the different approaches to serious web-based games for cognitive, motor, and sensory disabilities.

The third objective is to identify the WCAG-based accessibility guidelines applied to serious gaming to determine trends and gaps in serious game development.

The research questions were raised because serious games have been extensively incorporated into the teaching-learning processes [23,24]. Due to the increase in their use, the need arises to fully guarantee their accessibility to people with disabilities in educational environments.

Our study examines the results of existing primary studies published on accessibility and serious games to identify current trends and open issues in the domain: our research questions and each question's purpose.

**RQ1.** Are the web-based serious games being developed today accessible? To investigate the accessibility of web-based serious games that have been developed from 2000 to the present.

**RQ2.** What are the proposals to increase accessibility by disability in web-based serious games? To identify the accessibility proposals by the disability that are applied in web-based serious games.

**RQ3.** What are the accessibility solutions by disability for web-based serious games? To identify existing by disability solutions that are used to achieve accessibility in web-based serious games.

**RQ4.** What methods are applied in the design of web-based serious games? To classify the methods applied in the design of web-based serious games by disability.

**RQ5**. What kind of research and contributions are used in web-based serious games and accessibility? To identify the types of research and contribution used in web-based serious games and accessibility considering the disability.

*3.2. Search Strategy*

Primary studies were identified by a query string derived from the research questions. Based on the research questions in Table 1, PICO was implemented as follows:

- Population: published studies.
- Intervention: accessibility, web-based serious games.
- Comparison: selected studies by disability, accessibility standard-based, type of research, assistive technologies, and use of external devices.
- Outcome: published studies on accessibility and web-based serious games.

Built on PICO, we created the query string, as presented in Table 2. The search was conducted on 6 June 2020, and the authors selected four academic research databases used in engineering to retrieve primary information: (1) ACM Digital Library; (2) IEEE Xplore; (3) Scopus, and (4) Web of Science (WOS). The query strings for each chosen source were defined from the search terms connected by Boolean AND/OR operators. Additionally, the asterisk (*) wildcard was used to include both the singular and plural form of each term. Table 1 shows the selected database, the query string, and the number of studies extracted. The query string was applied to the title of the publication with the keywords: "serious", "game*", and "accessi*". Similar search syntax was applied across the four selected sources for consistency; the period under review included studies published between 2000 and 2020. We used equivalent strings that seek to locate the same articles, but each database has its specific syntax.

**Table 1.** Query string applied.

| Database | String Search | Number of Studies |
|---|---|---|
| **ACM Digital Library** | [Publication Title: accessi*] AND [Publication Title: serious] AND [Publication Date: (01/01/2000 TO 05/31/2020)]<br>[Publication Title: accessi*] AND [Publication Title: game*] AND [Publication Date: (01/01/2000 TO 05/31/2020)] | 92 |
| **IEEE Xplore** | ((“Document Title”: accessi* serious) OR “Document Title”: accessi* game*) | 25 |
| **Scopus** | TITLE (accessi*) AND (TITLE (serious) OR TITLE (game*)) | 190 |
| **Web of Science** | TI = (accessi* serious) OR TI = (accessi* game*) | 169 |
| **Total studies number** | | **476** |

**Table 2.** Article quality evaluation checklist.

| N° | Quality Assessment Questions | Answer |
|---|---|---|
| QA1 | Is serious games accessibility detailed in the paper? | (+1) Yes/(+0) No |
| QA2 | Is the serious games accessibility evaluation method specified in the paper? | (+1) Yes/(+0) No |
| QA3 | Does the paper discuss any findings of serious games accessibility evaluation? | (+1) Yes/(+0) No |
| QA4 | Are standard serious games accessibility errors described in the results? | (+1) Yes/(+0) No |
| QA5 | Is the journal or the conference where the paper was published indexed in SJR? | (+1) if it is ranked Q1, (+0.75) if it is ranked Q2, (+0.50) if it is ranked Q3, (+0.25) if it is ranked Q4, (+0.0) if it is not indexed. |

### 3.3. Screening of Documents

Based on the guidelines of the literature review [7], the application of the inclusion and exclusion criteria is essential to filter the results. The inclusion and exclusion criteria aim to obtain relevant primary studies to answer the defined research questions. Selection discrepancies were resolved by consensus between the authors.

Inclusion criteria: The primary study must be related to (1) the type of publication in journals, conferences, books, and book chapters on accessibility in web-based serious games published from 2000 to 2020; (2) primary peer-reviewed studies; (3) written in English language.

Exclusion criteria: The primary study: (1) summarizes a keynote, a workshop introduction, or only an abstract; (2) duplicate articles from the same study from different sources.

In this phase, to describe the process, we apply PRISMA [11,12,25], as shown in Figure 2. PRISMA applies to all types of systematic reviews and is not limited to clinical trials. PRISMA has been conceived as a tool to help improve clarity and transparency in systematic reviews. The search and selection process is placed on a flow chart; the process phases serve as a literature reviewer's guide. This process includes: (1) the databases consulted, indicating the number of documents obtained from each of them; (2) the number of documents that are duplicates; (3) the number of papers eliminated in each stage of the process and the reasons for elimination, and (4) the number of documents included in the study. Figure 3 shows the PRISMA flow diagram with the four phases of the selection process of the articles, which are described below:

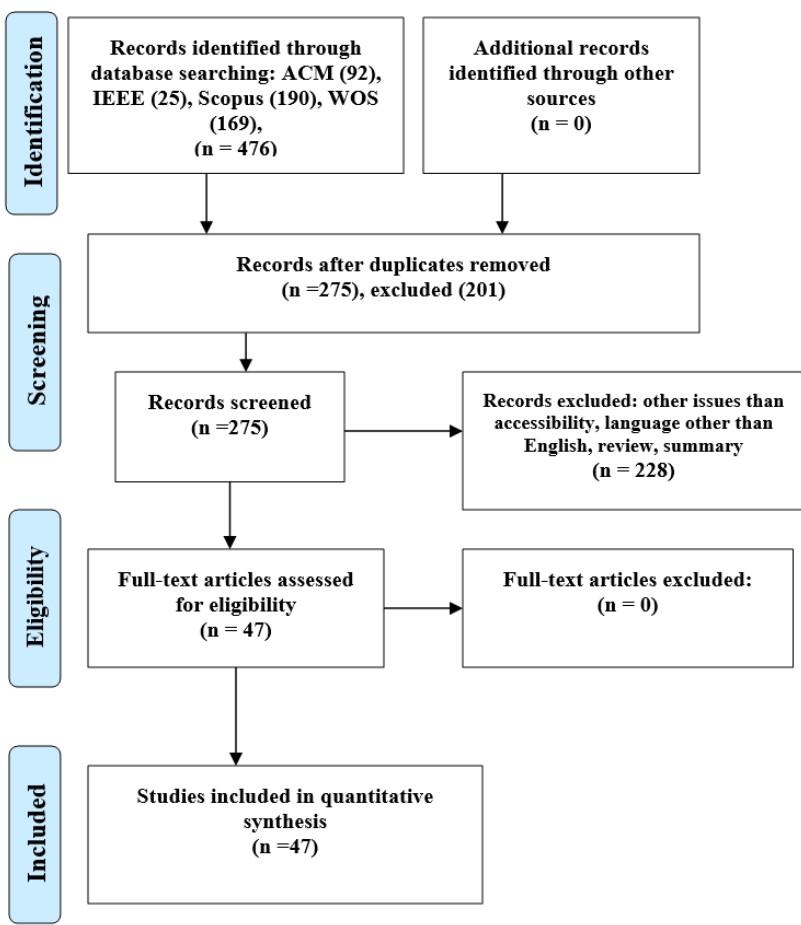

**Figure 3.** PRISMA flow diagram.

Phase 1: Identification. Here, we include the records obtained from database searches: ACM with 92 documents, IEEE Xplore with 25 articles, Scopus with 190 articles, and WOS with 169, a total of 476 articles were extracted.

Phase 2: Screening. Here, we apply the inclusion and exclusion criteria. Of the 476 articles, 201 articles were excluded because they were duplicated in different databases, 275 articles were included. Then, in the following filter, we excluded studies written in a language other than English, review studies, abstracts, workshops, and studies on topics other than accessibility in serious games; we excluded a total of 228 studies; finally, a total of 47 reviews passed to the next phase.

Phase 3: Eligibility. Three authors conducted an in-depth review of the full text of the 47 articles that focused explicitly on primary studies on accessibility and web-based serious games; we did not exclude any full-text articles. Furthermore, we evaluate the quality of the research articles that respond to accessibility in serious games; we apply a "quality assessment" of the selected articles. The purpose of this quality assessment (QA) is to weigh the importance of each of the papers chosen when the results are discussed and to guide the interpretation of findings [8]. Each QA obtains a score of one for the fulfillment of each clause (1) Is web-based serious games accessibility detailed in the paper? (2) Is the method of evaluating the accessibility of web-based games specified in the article? (3) Does the article discuss the accessibility assessment results in serious web-based games? (4) Are the accessibility issues of the web-based serious games described? (5) Is the journal where the paper was published indexed in SCImago Journal Rank (SJR)? Table 2 presents a checklist of quality evaluation.

Phase 4: Included. We recorded 47 articles full-text articles in the quantitative synthesis. Figure 3 shows that we have not added any additional records.

### 3.4. Keywording and Classification

For keywording and classification, we apply the concepts: (1) Classification scheme, which consists of a process of reading abstracts, searching for keywords and concepts that reflect the contribution of the primary study to ensure that the desired results are covered in the literature review, providing in the introduction a set of categories representing the study population. (2) Keywords that are used to apply the classification scheme in the literature review system. We consider abstract reading, keyword search, and context-related study objective. (3) The technique that allows for the classification of relevant articles for actual data extraction. In this phase, we use the keywords to group and form categories. The categories have been grouped. We review all the selected documents. After reading them, we update the categories or create a new category if the document reveals something. The keywords grouped by category, and the frequency are Key1 = Accessibility, Accessibility assessment, Accessibility, Design with a frequency of 37. Key2 = Assistive technology with 3. Key3 = Computer, aided, instruction with 10. Key4 = Disability with 5. Key5 = Games, Game, serious games, videogames with 44. Key6 = Guidelines, Accessibility guidelines with 14. Key7 = Human factors, Systems interaction with 5. Key8 = People disabilities with 25. Key9 = Visually impaired with 13. Key10 = Web with a frequency of 6.

### 3.5. Data Extraction

Data extraction was iterative; it was divided into several stages in which different activities were carried out. To extract the information from the ACM Digital Library, we exported it to BibTeX (BIB) format. In contrast, the IEEE Xplore, Scopus, and WOS information were exported in Research Information Systems (RIS) format. We then imported the data from the four files to the StartLapes tool version 2.3.4.2 [25], which automatically eliminates duplicate studies. We applied the process detailed in the PRISMA flowchart. Finally, we imported the data into a Microsoft Excel spreadsheet to continue the analysis. Table 3 presents the 47 primary studies selected, ordered by the most current year of publication. It contains the article number, the assigned indicator (it was created with the first letters of the surnames of the first two authors and the year of publication), the title of the article, the first author with the reference, and the year of publication.

In this phase, we apply the quality evaluation to the selected papers. Table 4 presents a list of the chosen works, together with the results of the quality control. Furthermore, a standardization column has been created to use a standard scale from 0 to 1.

**Table 3.** List of papers selected in this review.

| # | ID | Title | Authors | Year |
|---|---|---|---|---|
| 1 | RS20 | Be Active! Participatory Design of Accessible Movement-Based Games | Regal G [26] | 2020 |
| 2 | SD20 | A serious game to improve engagement with web accessibility guidelines | Spyridonis F [27] | 2020 |
| 3 | SA20a | Development of an accessible video game to improve the understanding of the test of Honey-Alonso | Salvador-Ullauri L [28] | 2020 |
| 4 | KO20 | Game accessibility and advocacy for participation of the Japanese disability community | Kaigo M [29] | 2020 |
| 5 | SA20b | Accessibility evaluation of video games for users with cognitive disabilities | Salvador-Ullauri L [20] | 2020 |
| 6 | KT19 | A Study on Accessibility in Games for the Visually Impaired | Khaliq I [30] | 2019 |
| 7 | DF19 | Startup Workplace, Mobile Games, and Older Adults: A Practical Guide on UX, Usability, and Accessibility Evaluation | De Lima Salgado A[31] | 2019 |
| 8 | OZ19 | Accessibility Requirements in Serious Games for Low Vision Children | Othman N [32] | 2019 |
| 9 | CP19a | Future design of accessibility in games: A design vocabulary | Cairns P [15] | 2019 |
| 10 | CM19 | A guide for making video games accessible to users with cerebral palsy | Compañ-Rosique P [33] | 2019 |
| 11 | SD19 | A Serious Game for Raising Designer Awareness of Web Accessibility Guidelines | Spyridonis F [34] | 2019 |
| 12 | CP19b | Enabled Players: The Value of Accessible Digital Games | Cairns P [35] | 2019 |
| 13 | MD19 | Problem-Based Learning applied to the development of accessible serious games | Martins V [36] | 2019 |
| 14 | JG18 | Towards an Accessible Mobile Serious Game for Electronic Engineering Students with Hearing Impairments | Jaramillo-Alcázar A [37] | 2018 |
| 15 | KK18 | Bonk: Accessible programming for accessible audio games | Kane S [38] | 2018 |
| 16 | JL18a | Accessibility assessment of serious mobile games for people with cognitive impairments | Jaramillo-Alcázar A [19] | 2018 |
| 17 | JL18b | An approach to mobile serious games accessibility assessment for people with hearing impairments | Jaramillo-Alcázar A [39] | 2018 |
| 18 | YC18 | Design of a game community based support system for cognitive game accessibility | Yildiz S [40] | 2018 |
| 19 | WK18 | Game Accessibility Guidelines and WCAG 2.0-A Gap Analysis | Westin T [41] | 2018 |
| 20 | WC18 | W3C accessibility guidelines for mobile games | Wilson A [42] | 2018 |
| 21 | LP17a | A Mobile Educational Game Accessible to All, Including Screen Reading Users on a Touch-Screen Device | Leporini B [43] | 2017 |
| 22 | SJ17 | A Serious Game Accessible to People with Visual Impairments | Salvador-Ullauri L [44] | 2017 |
| 23 | JL17 | Mobile Serious Games: An Accessibility Assessment for People with Visual Impairments | Jaramillo-Alcázar A [17] | 2017 |
| 24 | LP17b | An Inclusive Educational Game Usable via Screen Reader on a Touch-Screen | Leporini B [45] | 2017 |
| 25 | PC17 | Game Accessibility Guidelines for People with Sequelae from Macular Chorioretinitis | Pereira A [46] | 2017 |
| 26 | AF17 | Mobile audio games accessibility evaluation for users who are blind | Araújo M [47] | 2017 |
| 27 | LM16 | Interaction in Video Games for People with Impaired Visual Function: Improving Accessibility | López J [18] | 2016 |

**Table 3.** *Cont.*

| # | ID | Title | Authors | Year |
|---|---|---|---|---|
| 28 | HS16 | Using video game patterns to raise the intrinsic motivation to conduct accessibility evaluations | Henka A [48] | 2016 |
| 29 | DZ15 | Accessible Games for Blind Children, Empowered by Binaural Sound | Drossos K [49] | 2015 |
| 30 | WF15 | Games accessibility for deaf people: Evaluating integrated guidelines | Waki A [50] | 2015 |
| 31 | Po14 | Understanding and Addressing Real-World Accessibility Issues in Mainstream Video Games | Porter J R [51] | 2014 |
| 32 | MB14 | BraillePlay: Educational Smartphone Games for Blind Children | Milne L [52] | 2014 |
| 33 | TS14 | Development of a game engine for accessible web-based games | Torrente J [53] | 2014 |
| 34 | PK13 | Guidelines of Serious Game Accessibility for the Disabled | Park H [54] | 2013 |
| 35 | Ga13 | Game Accessibility: Enabling Everyone to Play | Garber L [55] | 2013 |
| 36 | WW13 | Return on investment in game accessibility for cognition impairments | Westin T [56] | 2013 |
| 37 | MM12 | Assessment of Universal Design Principles for Analyzing Computer Games' Accessibility | Mustaquim M [57] | 2012 |
| 38 | TV11 | Introducing accessibility features in an educational game authoring tool: The <e-adventure> experience | Torrente J [58] | 2011 |
| 39 | OM10 | Accessibility of a Social Network Game | Ossmann R [59] | 2010 |
| 40 | GS09 | Designing Universally Accessible Games | Grammenos D [60] | 2009 |
| 41 | MH08 | Accessibility Challenge—a Game Show Investigating the Accessibility of Computer Systems for Disabled People | Morgan M [61] | 2008 |
| 42 | MO08 | More than just a game: Accessibility in computer games | Miesenberger K [62] | 2008 |
| 43 | MP07 | Finger Dance: A Sound Game for Blind People | Miller D [63] | 2007 |
| 44 | OM06 | Guidelines for the development of accessible computer games | Ossmann R [64] | 2006 |
| 45 | GS06 | Access invaders: Developing a universally accessible action game | Grammenos D [65] | 2006 |
| 46 | OA06 | Computer Game Accessibility: From Specific Games to Accessible Games | Ossmann R [66] | 2006 |
| 47 | CL03 | The TiM game engine: Development of computer games accessible to blind and partially sighted children | Callaos N [67] | 2003 |

**Table 4.** Selected papers and quality assessment outcomes.

| ID | Publication Name | Quality Assessment | | | | | | |
|---|---|---|---|---|---|---|---|---|
| | | QA1 | QA2 | QA3 | QA4 | QA5 | Score | Normalization |
| RS20 | International Conference on Tangible, Embedded, and Embodied Interaction | 1 | 1 | 1 | 1 | 0 | 4 | 0.8 |
| SD20 | Behaviour & Information Technology | 1 | 1 | 1 | 1 | 0.75 | 4.75 | 0.95 |
| SA20a | International Conference on Applied Human Factors and Ergonomics | 1 | 1 | 1 | 1 | 0.5 | 4.5 | 0.9 |
| KO20 | Information | 1 | 1 | 1 | 1 | 0.5 | 4.5 | 0.9 |
| SA20b | International Conference on Intelligent Human Systems Integration | 1 | 1 | 1 | 1 | 0.5 | 4.5 | 0.9 |
| KT19 | International Conference on Smart Objects and Technologies for Social Good | 1 | 1 | 1 | 1 | 0 | 4 | 0.8 |
| DF19 | International Conference on Smart Objects and Technologies for Social Good | 1 | 1 | 1 | 1 | 0 | 4 | 0.8 |
| OZ19 | International Conference on the Design of Communication | 1 | 1 | 1 | 1 | 0 | 4 | 0.8 |
| CP19a | International Journal of Human Computer Studies | 1 | 1 | 1 | 1 | 1 | 5 | 1 |
| CM19 | Universal Access in the Information Society | 1 | 1 | 1 | 1 | 0.75 | 4.75 | 0.95 |
| SD19 | Conference on Human-Computer Interaction | 1 | 1 | 1 | 1 | 0 | 4 | 0.8 |
| CP19b | Games and Culture | 1 | 1 | 1 | 1 | 1 | 5 | 1 |
| MD19 | Iberian Conference on Information Systems and Technologies | 1 | 1 | 1 | 1 | 0.5 | 4.5 | 0.9 |
| JG18 | World Engineering Education Conference | 1 | 1 | 1 | 1 | 0 | 4 | 0.8 |
| KK18 | Conference on Interaction Design and Children | 1 | 1 | 1 | 1 | 0 | 4 | 0.8 |
| JL18a | International Conference on Information Systems and Computer Science | 1 | 1 | 1 | 1 | 0 | 4 | 0.8 |
| JL18b | International Conference on Information Theoretic Security | 1 | 1 | 1 | 1 | 0.5 | 4.5 | 0.9 |
| YC18 | International Conference on ArtsIT | 1 | 1 | 1 | 1 | 0.25 | 4.25 | 0.85 |
| WK18 | International Conference on Computers Helping People with Special Needs | 1 | 1 | 1 | 1 | 0.75 | 4.75 | 0.95 |
| WC18 | The Computer Games Journal | 1 | 1 | 1 | 1 | 0 | 4 | 0.8 |
| LP17a | World Conference on Mobile and Contextual Learning | 1 | 1 | 1 | 1 | 0 | 4 | 0.8 |
| SJ17 | International Conference on Education Technology and Computers | 1 | 1 | 1 | 1 | 0 | 4 | 0.8 |
| JL17 | International Conference on Technological Ecosystems for Enhancing Multiculturality | 1 | 1 | 1 | 1 | 0 | 4 | 0.8 |
| LP17b | ACM SIGACCESS Accessibility and Computing | 1 | 1 | 1 | 1 | 0 | 4 | 0.8 |
| PC17 | Entertainment Computing | 1 | 1 | 1 | 1 | 0.75 | 4.75 | 0.95 |
| AF17 | International Conference on Universal Access in Human-Computer Interaction | 1 | 1 | 1 | 1 | 0 | 4 | 0.8 |
| LM16 | International Conference on Human Computer Interaction | 1 | 1 | 1 | 1 | 0 | 4 | 0.8 |

**Table 4.** *Cont.*

| ID | Publication Name | Quality Assessment | | | | | | |
|----|------------------|------|------|------|------|------|-------|---------------|
| | | QA1 | QA2 | QA3 | QA4 | QA5 | Score | Normalization |
| HS16 | International Conference on Applied Human Factors and Ergonomics | 1 | 1 | 1 | 1 | 0.5 | 4.5 | 0.9 |
| DZ15 | International Conference on PErvasive Technologies Related to Assistive Environments | 1 | 1 | 1 | 1 | 0 | 4 | 0.8 |
| WF15 | International Conference on Universal Access in Human-Computer Interaction | 1 | 1 | 1 | 1 | 0 | 4 | 0.8 |
| Po14 | ACM SIGACCESS Accessibility and Computing | 1 | 1 | 1 | 1 | 0 | 4 | 0.8 |
| MB14 | International ACM SIGACCESS Conference on Computers & Accessibility | 1 | 1 | 1 | 1 | 0 | 4 | 0.8 |
| TS14 | International Conference on Games and Learning Alliance | 1 | 1 | 1 | 1 | 0.75 | 4.75 | 0.95 |
| PK13 | International Conference on Information Science and Applications | 1 | 1 | 1 | 1 | 0 | 4 | 0.8 |
| Ga13 | Computer | 1 | 1 | 1 | 1 | 1 | 5 | 1 |
| WW13 | European Conference of the Association for the Advancement of Assistive Technology in Europe | 1 | 1 | 1 | 1 | 0 | 4 | 0.8 |
| MM12 | International Conference on Computers for Handicapped Persons | 1 | 1 | 1 | 1 | 0.75 | 4.75 | 0.95 |
| TV11 | International Conference on Advanced Learning Technologies | 1 | 1 | 1 | 1 | 0 | 4 | 0.8 |
| OM10 | International Conference on Computers for Handicapped Persons | 1 | 1 | 1 | 1 | 0.75 | 4.75 | 0.95 |
| GS09 | Computers in Entertainment | 1 | 1 | 1 | 1 | 0.5 | 4.5 | 0.9 |
| MH08 | ACM SIGACCESS Accessibility and Computing | 1 | 1 | 1 | 1 | 0 | 4 | 0.8 |
| MO08 | Symposium of the Austrian HCI and usability engineering group | 1 | 1 | 1 | 1 | 0 | 4 | 0.8 |
| MP07 | International ACM SIGACCESS Conference on Computers and Accessibility | 1 | 1 | 1 | 1 | 0 | 4 | 0.8 |
| OM06 | International Conference on Computers for Handicapped Persons | 1 | 1 | 1 | 1 | 0 | 4 | 0.8 |
| GS06 | International Conference on Computers for Handicapped Persons | 1 | 1 | 1 | 1 | 0 | 4 | 0.8 |
| OA06 | International Conference on Computer Games (CGAMES) | 1 | 1 | 1 | 1 | 0 | 4 | 0.8 |
| CL03 | World Multiconference on Systemics, Cybernetics and Informatics | 1 | 1 | 1 | 1 | 0 | 4 | 0.8 |

Normalization [68] was used, which preserves the relationship between the original data values. The values in this column are transformed using the following Equation (1):

$$Normalization = \frac{Score - \min(Score)}{[\max(Score) - \min(Score)]} \tag{1}$$

where the min(Score) is equal to 0, the max(Score) is equal to 5, and the Score is the value calculated in Table 4.

## 4. Results

In this section, we answer the research questions by:

(1)   A bibliometric analysis to collect information about the authors and publication data of research growth over time, journals, conferences, books, and book chapters published on serious games and accessibility.

(2)   A literature review to map the studies according to serious games' concepts and the five research questions.

### 4.1. Bibliometric Analysis

This analysis aims to respond to RQ1; Figure 4 shows the evolution of scientific production, presenting the number of documents each year. The years of most scientific output in accessibility in the serious games are 2018 and 2019. We found eight papers for 2019, which corresponds to 17%, seven documents for 2018, which corresponds to 14.9%. In 2017, we found six articles, corresponding to 12.8%. In 2020, we found five documents, which corresponds to 10.6%. It is expected that this number tends to increase because it was done until July 2020. In 2006, 2013, and 2014, there were three documents each year that add up to 19.1%. In 2008, 2015, and 2016, there were two items each year that add up to 12.8%. Finally, in 2003, 2007, 2009, 2010, 2011, and 2012, one document added up to 12.8% each year. The annual growth rate of the published articles follows the polynomial Equation (2).

$$y = 0.0216x2 - 0.1639x + 0.6421 \tag{2}$$

Figure 5 presents 35 conference studies representing 74.5% of the total and 12 journal articles representing 25.5%. In this review of the literature, the most significant number of studies found are concentrated in conferences. The largest number of documents are indexed in Scopus.

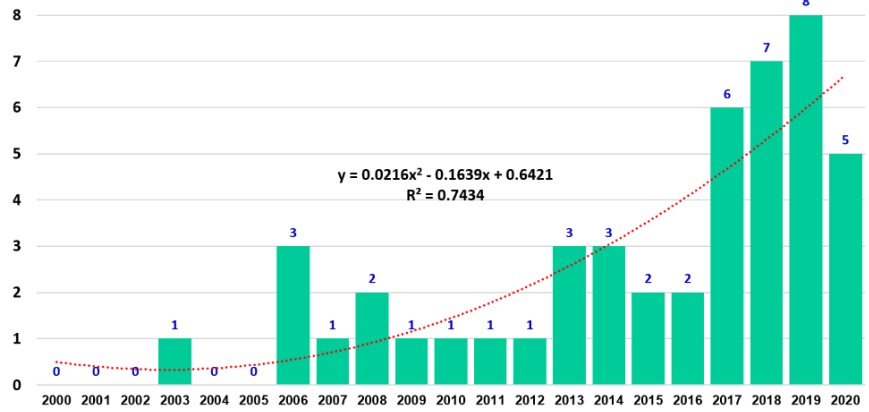

**Figure 4.** Documents published from 2000 to 2020.

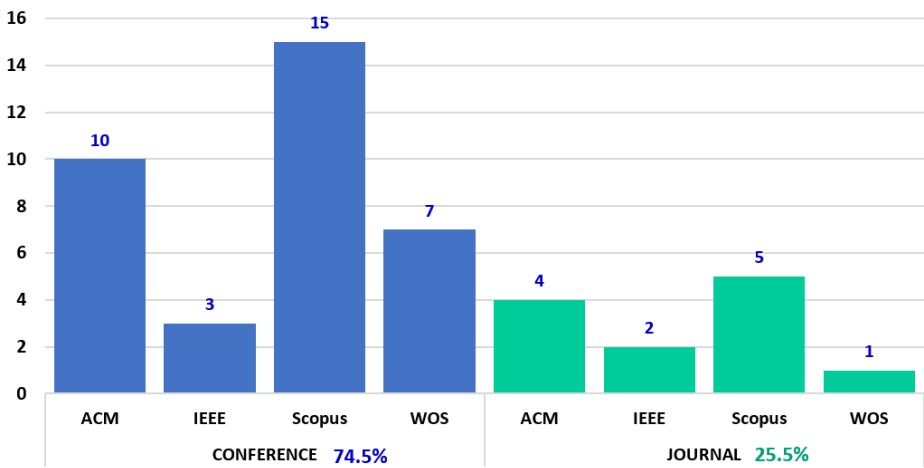

**Figure 5.** Documents by type.

## 4.2. Review of the Literature to Map the Studies

In this section, we presented a classification scheme using keywords; we performed the following: (1) we read the abstracts of the 47 primary studies selected and searched for keywords;(2) we read the introduction and conclusion sections of each of the primary studies chosen to elaborate the classification scheme; (3) we presented the classification in five aspects that were defined as follows: (1) The disability accessibility guidelines which aim to provide a standard of easy access to serious games that meets people's needs. (2) The applied solutions that involve the methods used to make serious games accessible. (3) The disability accessibility guidelines that provide input for serious play. (4) The types of disability contributions [5] include the formal study, method, system, or experience, and (5) The type of research that includes validation, solution, evaluation, feedback, and experience [7].

Then, we present and discuss the answers to the research questions in this study; the dataset and analysis are available for replication in the Mendeley repository [69].

### 4.2.1. RQ1. Are the Serious Games Being Developed Today Accessible?

In this research, we have selected the primary studies that apply accessibility to serious games. The web has numerous limitations, but if the design is considered accessible to all people, including people with some type of disability, the application will be more accessible and inclusive for many users.

Table 5 presents the primary studies that use accessibility by type of disability. In the review of the documents, we applied the following definitions:

- Cognitive or intellectual disability is a problem characterized by a delay in mental development that disrupts the learning process.
- Motor coordination, or physical disability, is a problem related to significant impairment of one or more parts of the body's movement abilities.
- Sensory: this type of disability is related to (1) vision, which includes users with low vision and deafness; (2) hearing disability, which provides deafness and hearing loss.

Table 5 shows the accessibility related to the type of disability. We found 33 primary studies on sensory disability that represent 72% of the total, then 11 studies apply accessibility for cognitive disability with 22%. Finally, three studies on motor coordination that correspond to 6% of the total.

**Table 5.** Studies by disability.

| Type of Disability | ID |
|---|---|
| Cognitive (11 studies) | CL03, CM19, HS16, JL18a, LP17a, MD19, OM06, SA20a, SA20b, WW13, YC18 |
| Motor coordination (3 studies) | KO20, DF19, SD19 |
| Sensory: visually impaired, hearing (33 studies) | JL17, JG18, RS20, SD20, KT19, OZ19, CP19a, CP19b, KK18, JL18b, WK18, WC18, SJ17, LP17b, PC17, AF17, LM16, DZ15, WF15, Po14, MB14, TS14, PK13, Ga13, MM12, TV11, OM10, GS09, MH08, MO08, MP07, GS06, OA06 |

### 4.2.2. RQ2. What Are the Proposals to Increase Accessibility in Serious Games?

Table 6 presents the proposals to increase accessibility by type of disability; in the review of the documents, we applied the following definitions:

- WCAG 2.0: includes the primary studies that applied the Web Content Accessibility Guidelines with version 2.0 to increase serious games accessibility.
- WCAG 2.1: includes the primary studies that applied the Web Content Accessibility Guidelines with version 2.1 to increase serious games accessibility. WCAG 2.1 is the most advanced and accepted mechanism for creating accessible content, and it is not limited exclusively to web content [70].
- Other guidelines contain the primary studies that help increase accessibility in serious games by applying guidelines without specifying the standard.
- External devices: contains the primary studies that help to increase accessibility use some form of support appropriate to the motor, cognitive, and sensory characteristics of people who find it easy to access serious play, including the use of assistive technology.

We found: (1) 24 studies that refer to the guidelines for achieving accessibility in serious games but do not specify a standard, representing 51.1% of the total; (2) 14 studies that indicate external devices' application to achieve a higher level of accessibility in serious games, representing 29.8%; (3) seven studies that focused on the guidelines suggested by WCAG 2.0, representing 14.9%; (4) two studies focused on WCAG 2.1, representing 4.3% of the total.

**Table 6.** Guidelines by disability.

| Guidelines | Cognitive | Motor Coordination | Sensory (Visually Impaired, Hearing) |
|---|---|---|---|
| WCAG 2.0 (7 studies) | SA20a. | | SD20, CP19a, WK18, AF17, WF15, PK13. |
| WCAG 2.1 (2 studies) | SA20b. | | WC18. |
| Other guidelines (24 studies) | CL03, CM19, LP17a, MD19, OM06. | KO20, DF19, SD19. | JG18, RS20, KT19, OZ19, CP19b, KK18, PC17, DZ15, MB14, MM12, TV11, GS09, MH08, MP07, GS06, OA06. |
| External devices (14 studies) | HS16, JL18a, WW13, YC18. | | JL17, JL18b, SJ17, LP17b, LM16, Po14, TS14, Ga13, OM10, MO08. |

### 4.2.3. RQ3. What Are the Accessibility Solutions Proposed for Serious Games?

Table 7 presents seven accessibility solutions for serious games by disability; it contains the type of solution applied and references; in the review of the documents, we used the following definitions:

- Accessibility guidelines: contains the primary studies that give solutions to accessibility problems by applying some norm or standard.
- Accessibility requirements: includes primary studies that provide solutions to accessibility problems by suggesting examining the state of serious games.
- Apply assistive technologies contains studies in which they apply assistive technology to achieve accessibility in serious games.

- Apply the concept of parallel game: has primary studies in which they give accessibility solutions by applying the parallel game, parallel universes, or alternative realities that improve the experience by taking the player out of the reality they are used to, and helps them improve their concentration.
- Apply external devices: contain studies of adaptation of external devices to provide accessibility solutions for serious games.
- Companies games: include primary studies where accessibility depends on the company's standards that develop serious games.
- Creative design: contains studies where applying innovative design addresses some of the accessibility issues.

The solution by type of disability has: (1) 22 studies and corresponds to 46.8% of the total consists of accessibility guidelines; (2) external devices apply with 14 primary reviews, corresponding to 29.8% of the total; (3) the solution proposed consists of accessibility requirements, with four studies representing 8.5%; (4) the proposal to apply creative design, with three primary studies, represents 6.4%; (5) the solution consists of applying the concept of the parallel game with two studies, representing 4.3%. (6) the proposal indicates the application of assistive technologies, has one study, corresponding to 2.1%; (7) the solution that shows to interact with companies games, with one review, representing 2.1% of the total.

**Table 7.** The solution to accessibility by type of disability.

| Solution | Cognitive | Motor Coordination | Sensory (Visually Impaired, Hearing) |
|---|---|---|---|
| Accessibility guidelines (22 studies) | SA20a, SA20b, MD19, LP17a, CL03. | DF19, SD19. | JG18, KT19, CP19b, WK18, WC18, PC17, DZ15, WF15, MB14, PK13, MM12, TV11, MH08, MP07, OA06. |
| Accessibility requirements (4 studies) | CM19 | | OZ19, CP19a, AF17. |
| Apply assistive technologies (1 study) | OM06 | | |
| Apply the concept of Parallel Game (2 studies) | | | GS09, GS06. |
| Apply external devices (14 studies) | JL18a, YC18, HS16, WW13. | | JL17, JL18b, SJ17, LP17b, LM16, Po14, TS14, Ga13, OM10, MO08. |
| Companies Games (1 study) | | KO20 | |
| Creative design (3 studies) | | | RS20, SD20, KK18. |

### 4.2.4. RQ4. What Methods Are Applied in the Design of Serious Games?

Table 8 presents a summary of the methods by disability applied in the design of serious games with the references of the selected studies; in the review of the documents, we used the following definitions:

- Qualitative: The qualitative method is inductive and follows a flexible design, and records are made through narration, participant observation, and unstructured interviews. This method is manifested in the facts, processes, observations, case studies, interviews, analysis, and opinions of the authors are very subjective because there is no measurement of the elements. This method includes small-scale studies, emphasizes the validity of research through proximity to empirical reality, and does not usually test theories or hypotheses. The basis of this method is intuition; in general, it does not allow statistical analysis.
- Quantitative: The quantitative method produces numerical data, which allows the data to be collected and analyzed. In this method, objectivity is the way to reach knowledge; it uses specific and controlled measurements, looking for certainty. It includes descriptive studies under the

objective conception through a deductive strategy. This method contains studies that apply mixed methods and surveys for data collection.

Table 8 contains 41 studies that apply the qualitative method, representing 87.2%, and six studies use the quantitative approach, corresponding to 12.8% of the total. Figure 6 presents a classification of the studies by the method applied. The qualitative studies include (1) two analytical studies, corresponding to 4.3% of the total; (2) four case studies, representing 8.5%; (3) one interview corresponds to 2.1%; (4) 28 observation studies, corresponding to 59.6% of the total, are the most significant number of studies in this literature review; (5) three opinion studies compared to 6.4%. Quantitative studies include (1) two studies applying mixed methods represent 4.3% and (2) seven studies based on surveys, corresponding to 14.9% of the total.

**Table 8.** Methods applied to the design of serious games by disability.

| Method | Cognitive | Motor Coordination | Sensory (Visually Impaired, Hearing) |
|---|---|---|---|
| Qualitative (41 studies) | OM06, SA20b, CM19, MD19, JL18a, YC18, LP17a, WW13, CL03. | KO20, DF19, SD19. | JL17, JG18, RS20, KT19, CP19a, KK18, JL18b, WC18, SJ17, LP17b, PC17, AF17, LM16, DZ15, WF15, Po14, MB14, TS14, PK13, Ga13, MM12, TV11, OM10, GS09, MH08, MO08, MP07, GS06, OA06. |
| Quantitative (6 studies) | SA20a, HS16 | | SD20, OZ19, CP19b, WK18. |

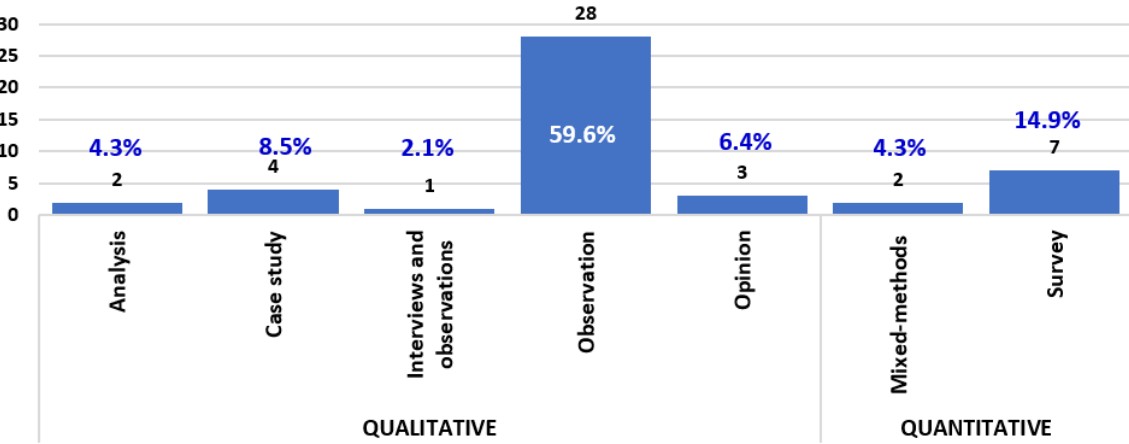

**Figure 6.** Methods and techniques applied in the design of serious games.

### 4.2.5. RQ5. What Kind of Research and Contribution Are Used in Accessibility in Serious Games?

To answer this question, we reviewed the documents by type of research by disability, as shown in Table 9. In the review of the papers, we applied the following definitions [7]:

- Evaluation research: provides the implemented solution, and investigates a practical problem, applies math test, survey, case study, field experiment to validate knowledge affirmation.
- Experience: contains case studies, projects, or reports on experiences, provides lessons learned.
- Opinion paper: provides the author's opinion on how something should be done.
- Solution proposal: offers a novel technique, or at least a relevant improvement.
- Validation research: presents researched techniques that have not yet been implemented in practice and are novel. It is methodologically sound and comprehensive, including experiments, prototyping, property testing, and simulation.

We found (1) 37 studies that apply "Experience", representing 78.7% of the total. (2) Five studies involve "Opinion paper" with 10.6%. (3) Two studies use "Evaluation research" corresponding to 4.3%.

(4) Two studies use "Validation research" with 4.3%. (5) One study applies a "Solution proposal" that represents 2.1% of the total. In this study, we conducted a systematic review of the literature with the following: (1) The question most answered corresponds to RQ1 with 47 studies, representing 28%. (2) The question RQ4, with 38 studies, represents 22.6%. (3) Question RQ5 has 37 studies, corresponding to 22%. (4) Questions RQ2 and RQ3 have 23 studies each, accounting for 27.4% of the total.

**Table 9.** Studies according to the type of research by disability.

| Research Type | Cognitive | Motor Coordination | Sensory (Visually Impaired, Hearing) |
|---|---|---|---|
| Evaluation research (2 studies) | | | CP19b, WK18. |
| Experience (37 studies) | OM06, SA20b, CM19, MD19, JL18a, YC18, LP17a, WW13, CL03. | DF19, SD19. | JL17, JG18, RS20, SD20, KT19, OZ19, CP19a, KK18, JL18b, WC18, SJ17, LP17b, PC17, AF17, LM16, DZ15, WF15, Po14, MB14, TS14, PK13, TV11, OM10, GS09, MP07, GS06. |
| Opinion paper (5 studies) | | | Ga13, MM12, MH08, MO08, OA06. |
| Solution proposal (1 study) | SA20a. | | |
| Validation research (2 studies) | HS16. | KO20. | |

Figure 7 shows (1) The question that contributes most to this study are RQ1, RQ2, and RQ3. (2) The type of research that contributes the most to this study is the experience, which contributes to the five research questions, corresponding to 27.8% of the total. (3) Evaluation research, solution proposal, and validation research contribute to RQ1, RQ2, and RQ3, representing 50%. (4) Opinion paper contributes to questions RQ1, RQ2, RQ3, and RQ4, corresponding to 22.2% of the total.

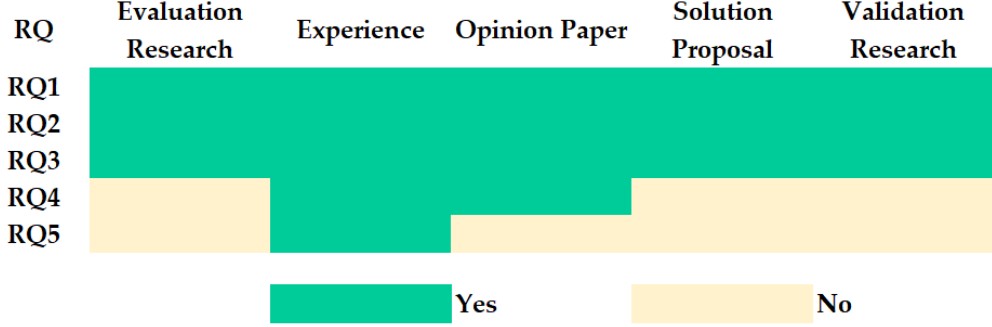

**Figure 7.** Contribution of research questions to the study.

Figure 8 shows: (1) A summary of the ten most frequently repeated keywords in the primary studies examined. The most repeated keyword is "game and serious games" with 27.2%, followed by "accessibility" corresponding 22.8%, "people disabilities" representing 15.4%, "guidelines" with 8.6%, "visually impaired" reaching 8%, and the rest representing 17.9%. (2) A summary of the answers to the questions posed in this research, and we found 17 studies that respond to the five questions and are AF17, CP19a, JL18a, JL18b, JL17, LP17b, LM16, OM10, PK13, Po14, KO20, SJ17, TS14, WF15, WW13, WC18, YC18.

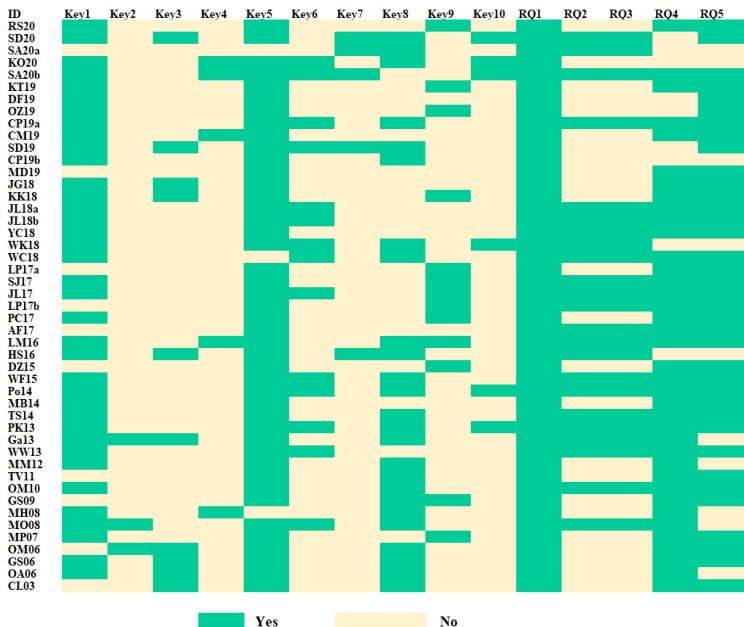

**Figure 8.** Primary studies answered keywords and research questions.

## 5. Discussion

In this study, we conducted a systematic review of the literature with five research questions. Excel spreadsheets can be downloaded from the Mendeley repository [69]. In this section, we summarize the main findings.

(1)  The current accessibility situation shows that few developers apply it to serious games that consider disabilities. This literature review found that 72% of the primary studies selected are related to visual and hearing impairment accessibility. On the other hand, 22% of studies apply accessibility guidelines to solve some cognitive and learning problems. Few studies, about 6%, are concerned with using accessibility to serious games for motor disabilities.

(2)  This study's statistics related to proposals to increase accessibility by type of disability in serious games reveal that 70.2% apply the accessibility guidelines. However, no guidelines are specified; we found studies that use their guidelines or modifications based on the International Game Developers Association (IGDA) [71] ideas on games' accessibility. Others apply the GA-SIG guidelines to create something like the WCAG [6]. We also find studies that involve the IBM [72] and Section 508 [73] guidelines. Of the selected primary studies, 14.9% use the WCAG 2.0, and 4.3% apply the WCAG 2.1. They apply the GA-SIG guidelines to create something like the W3C/WAI [74].

(3)  The solutions to increase accessibility in serious games considering disability are few; we found that 2.1% of assistive technologies are applied, especially for cognitive disabilities. Moreover, 29.8% use external devices to make games more accessible, especially for motor and sensory disabilities. According to the studies found, we can conclude no inclusive development culture in software development companies.

(4)  The methods applied to the design of serious disability games show that 87.2% of the primary studies selected use the qualitative approach, relying on observation to collect non-numerical data through focus groups and observation techniques. In contrast, the quantitative method received 12.8% of the selected primary studies, based on systematic empirical research of observable phenomena using statistical, mathematical, or computer techniques.

(5)  The SLR results by type of disability-related documents reveal that experience-based research models received the most attention in 78.7% of the selected primary studies. This research model includes case studies, projects, and experience reports that provide lessons learned.

Besides, 10.6% of the selected primary studies rely on the author's opinion to apply accessibility in serious games. 4.3% of selected primary studies rely on evaluation research to provide solutions to practical problems and surveys. Also, 4.3% of the primary studies apply validation research by presenting researched techniques that have not yet been implemented and are novel. Finally, 2.3% of the selected primary studies use solution proposals to offer new approaches and improvements.

The selected studies' quality was determined by applying the quality assessment based on five additional questions (see Table 5). This SLR process has its limitations; it is not foolproof as any other secondary research method.

In this study, attention was paid to choosing the most useful query strings adapted according to each database's query structure. To mitigate this limitation, we applied the PICO criteria to our query strings [5]. The selection of the databases, the ACM Digital Library, IEEE Xplore, Scopus, and WOS, is adequate because out of 476 articles, 201 were duplicates, which means that the coverage of the four databases is high, so much so that some of them could have been excluded. The criteria emerged from discussions with the researchers involved. However, primary research search terms, such as accessibility and serious games, are traditional, well-defined, and accepted terms, which should decrease the number of ignored studies. Moreover, as the study focuses on identifying primary research on accessibility and web-based serious games, there is less concern about capturing vaguely domain-related research.

## 6. Conclusions

Accessibility is an essential research area that emerges from the web. In recent years, many studies have been published with a growing interest in this topic. This study highlighted current trends and outstanding issues in accessibility and applied guidelines for designing serious inclusive games using the results of existing primary studies published between 2000 and 2020.

In this study, an SLR was conducted with a set of five research questions and five questions to validate the quality of the selected studies. We extracted a total of 476 studies, and after a screening process with the help of the PRISMA flowchart, we chose a group of 47 primary studies. As a result, the limitations and problems of serious games regarding accessibility and possible solutions to generate more inclusive serious games were demonstrated.

Furthermore, we identified the status of serious gaming and accessibility related to disability. We identified research and contribution types that apply to serious gaming in cognitive, motor, and sensory disabilities. This study identified that developers rely on assistive technologies through software and hardware to achieve greater accessibility in serious games. This study shows the need for research on issues related to accessibility policies, guidelines, and practices for serious games and the threats of accessibility violations. Moreover, this study shows open research issues in applying accessibility guidelines in serious games by companies and developers. Finally, this study provides researchers and professionals with the status of serious games related to cognitive, motor, and sensory disabilities. For future work, we suggest: (1) building a software tool that applies WCAG 2.1 guidelines to support serious game developers; (2) defining anti-rules to increase the accessibility of serious games; (3) conducting a literature review on the accessibility of serious games for mobile and computer applications for users with disabilities.

**Author Contributions:** Conceptualization, L.S.-U., and P.A.-V.; methodology, L.S.-U.; investigation, L.S.-U., and P.A.-V.; writing—original draft preparation, L.S.-U., and P.A.-V.; writing—review and editing L.S.-U., S.L.-M., and P.A.-V.; supervision, S.L.-M.; project administration, P.A.-V., and L.S.-U. All authors have read and agreed to the published version of the manuscript.

**Funding:** This research was funded by Universidad de Las Américas-Ecuador, as part of an internal research project FGE.PAV.19.11.

**Conflicts of Interest:** The authors declare no conflict of interest.

# Appendix A

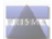 **PRISMA 2009 Checklist**

| Section/topic | # | Checklist item | Reported on page # |
|---|---|---|---|
| **TITLE** | | | |
| Title | 1 | Identify the report as a systematic review, meta-analysis, or both. | 1 |
| **ABSTRACT** | | | |
| Structured summary | 2 | Provide a structured summary including, as applicable: background; objectives; data sources; study eligibility criteria, participants, and interventions; study appraisal and synthesis methods; results; limitations; conclusions and implications of key findings; systematic review registration number. | 1,2 |
| **INTRODUCTION** | | | |
| Rationale | 3 | Describe the rationale for the review in the context of what is already known. | 1 |
| Objectives | 4 | Provide an explicit statement of questions being addressed with reference to participants, interventions, comparisons, outcomes, and study design (PICOS). | 1, 3 |
| **METHODS** | | | |
| Protocol and registration | 5 | Indicate if a review protocol exists, if and where it can be accessed (e.g., Web address), and, if available, provide registration information including registration number. | 3 |
| Eligibility criteria | 6 | Specify study characteristics (e.g., PICOS, length of follow-up), and report characteristics (e.g., years considered, language, publication status) used as criteria for eligibility, giving rationale. | 4 |
| Information sources | 7 | Describe all information sources (e.g., databases with dates of coverage, contact with study authors to identify additional studies) in the search, and date last searched. | 4 |
| Search | 8 | Present full electronic search strategy for at least one database, including any limits used, such that it could be repeated. | 4 |
| Study selection | 9 | State the process for selecting studies (i.e., screening, eligibility, included in systematic review, and, if applicable, included in the meta-analysis). | 4 |
| Data collection process | 10 | Describe the data extraction method from reports (e.g., piloted forms, independently, duplicate) and any processes for obtaining and confirming data from investigators. | 6,7 |
| Data items | 11 | List and define all variables for which data were sought (e.g., PICOS, funding sources) and any assumptions and simplifications made. | 3 |
| Risk of bias in individual studies | 12 | Describe methods used for assessing the risk of bias of individual studies (including specification of whether this was done at the study or outcome level) and how it is used in any data synthesis. | 3 |
| Summary measures | 13 | State the principal summary measures (e.g., risk ratio, the difference in means). | 13,14 |
| Synthesis of results | 14 | Describe the methods of handling data and combining results of studies, if done, including measures of consistency (e.g., $I^2$) for each meta-analysis. | 9,10 |
| Risk of bias across studies | 15 | Specify any assessment of the risk of bias that may affect the cumulative evidence (e.g., publication bias, selective reporting within studies). | 16 |
| Additional analyses | 16 | Describe methods of additional analyses (e.g., sensitivity or subgroup analyses, meta-regression), if done, indicating which were pre-specified. | 3-15 |

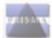 **PRISMA 2009 Checklist**

| Section/topic | # | Checklist item | Reported on page # |
|---|---|---|---|
| **RESULTS** | | | |
| Study selection | 17 | Give numbers of studies screened, assessed for eligibility, and included in the review, with reasons for exclusions at each stage, ideally with a flow diagram. | 7,8 |
| Study characteristics | 18 | For each study, present characteristics for which data were extracted (e.g., study size, PICOS, follow-up period) and provide the citations. | 9 |
| Risk of bias within studies | 19 | Present data on the risk of bias of each study and, if available, any outcome level assessment (see item 12). | 17,18 |
| Results of individual studies | 20 | For all outcomes considered (benefits or harms), present, for each study: (a) simple summary data for each intervention group (b) effect estimates and confidence intervals, ideally with a forest plot. | 4-15 |
| Synthesis of results | 21 | Present results of each meta-analysis done, including confidence intervals and measures of consistency. | 14,15 |
| Risk of bias across studies | 22 | Present results of any assessment of the risk of bias across studies (see Item 15). | 16 |
| Additional analysis | 23 | Give results of additional analyses, if done (e.g., sensitivity or subgroup analyses, meta-regression [see Item 16]). | 17 |
| **DISCUSSION** | | | |
| Summary of evidence | 24 | Summarize the main findings, including the strength of evidence for each main outcome; consider their relevance to key groups (e.g., healthcare providers, users, and policymakers). | 13-15 |
| Limitations | 25 | Discuss limitations at study and outcome level (e.g., risk of bias) and at review-level (e.g., incomplete retrieval of identified research, reporting bias). | 16,17 |
| Conclusions | 26 | Provide a general interpretation of the results in the context of other evidence and implications for future research. | 16,17 |
| **FUNDING** | | | |
| Funding | 27 | Describe funding sources for the systematic review and other support (e.g., a supply of data) and funders' role for the systematic review. | 17 |

*From:* Moher D, Liberati A, Tetzlaff J, Altman DG, The PRISMA Group (2009). Preferred Reporting Items for Systematic Reviews and Meta-Analyses: The PRISMA Statement. PLoS Med 6(7): e1000097. doi:10.1371/journal.pmed1000097

For more information, visit: www.prisma-statement.org

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
