# Peer review of "Web-Based Serious Games and Accessibility: A Systematic Literature Review"

_applsci, doi:10.3390/app10217859_

Round 1

Reviewer 1 Report

Although the enumeration appears in the citations, the authors' surnames are also shown (example line 191; 200; 205; 210; 213; 218;…)

It should be clarified in more detail why the PRISMA method is used for the review, if a typical protocol in the area of knowledge of Health Sciences (Medicine and Nursing).

Since the judgment of the evaluator, I think it would be more clear and legible if the results of the discussion are separated. In this sense, the sections of point 4 complicate the organization and logical reading of the article. On the other hand, in point 4 “Results and discussion”, the discussion is not appreciated, only the presentation of general and key results.

Author Response

Although the enumeration appears in the citations, the authors' surnames are also shown (example line 191; 200; 205; 210; 213; 218;…)

Dear reviewer, thank you so much for your constructive feedbacks and review. We have followed your suggestions and updated the manuscript.

Several studies [15], [16] show a lack of commitment by designers to implement accessibility. For this reason, there is a low percentage of accessible serious games.

Currently, serious games have been incorporated into the teaching-learning processes; therefore, it is essential to guarantee accessibility [17] so that the largest number of people can use them.

The article [18] presents an analysis of the accessibility guidelines for the development of videogames; the study is oriented to cognitive disabilities. It also proposes categorizing the guidelines that should be used to analyze a video game's accessibility, especially serious games. The authors present an evaluation tool for the development of serious games for mobile devices.

Following the article [19], the authors indicate that it is a great challenge to implement serious

It should be clarified in more detail why the PRISMA method is used for the review if a typical protocol in the area of knowledge of Health Sciences (Medicine and Nursing).

Dear reviewer, thank you very much for your comment. PRISMA is effectively applied in health systems, but its application is not limited. So we have adapted PRISMA and added the PRISMA checklist (see Appendix A), so we can see the adaptation we applied to our literature review study.

Since the judgment of the evaluator, I think it would be more clear and legible if the results of the discussion are separated. In this sense, the sections of point 4 complicate the organization and logical reading of the article. On the other hand, in point 4 “Results and discussion”, the discussion is not appreciated, only the presentation of general and key results.

Thank you very much for your valuable feedback. We have restructured the manuscript and separated the sections into results and discussion (section 4 and section 5).

Results

 In this section, we answer the research questions by:

 1) A bibliometric analysis to collect information about the authors and publication data of research growth over time, journals, conferences, books, and book chapters published on serious games and accessibility.

2) A literature review to map the studies according to serious games' concepts and the five research questions.

Discussion

In this study, we conducted a systematic review of the literature with five research questions. Excel spreadsheets can be downloaded from the Mendeley repository [68]; in this section, we summarize the main findings.

Reviewer 2 Report

The article describes a structured literature review on accessibility in serious games. 4 frequently used literature databases are searched, at the end there are 50 articles left, which are classified more detailed. The classification is used to answer 5 research questions.

The article is focused on a research topic worthy of investigation and comes to relevant results. Before publication, however, some issues should be addressed:

- The article is very long, because besides the actual research questions many general remarks about SLR are made, which however do not advance the actual research questions. Here a focus should be made and all explanations that are not directly necessary to answer the research questions should be removed from the article. For example, there is no need to explain the advantages and disadvantages of SLR in general terms, unless they are directly related to the research question.

- Basically the article should be checked once more for a logical structure. Some explanations are doubled, for example in section 1 and section 2, some explanations do not fit into the respective chapter. As a rule, all facts should be given only once. This also includes the information given in tables and graphics, some of which are redundant. Details about these two remarks can also be found in the further remarks.

- The reasons for the selection of research questions should be discussed.

- The results found and their implications should be better discussed. Which effects have which results? What other questions arise from these findings? What do these results imply for my own development of a serious game?

- Sometimes the results are only described with descriptive statistics. But it would be good to pick out examples to give the reader an impression of the typical state of accessibility in serious games. So in the end, all that remains is the impression that there are two standards, but otherwise little is done and if so, own solutions or special hardware is used.  This impression needs to be extended.

- The article should be optimized for better readability. This includes partly joingen sections (a section has more than one sentence), the use of bullet points, the addition of connection sentences between sections or subsections, the use of bold print etc.

General remarks

  • The abstract does not provide a definition for accessibility. It starts with a definition for serious games and describes thereafter the methodology used for the literature review. It fails to motivate pursuing the literature research.
  • What are the rationales for the research questions? Why this five and not another six research questions? It would be beneficial to discuss this a bit.
  • Section 1:
    • The definition of serious games in the abstract differs from the definition in the main text. Please be consistent.
    • A bit lengthy. For example, the sense and purpose of a literature review does not need to be explained (L71).
    • 102 “to verify if any gaps could be filled with the SLR proposed in this paper.” An SLR is proposed? It is done!? Further, this seems to be part of the rational of the entire study – so, why is this not presented earlier in the paper?
  • Section 2
    • Section „2.2. Disability and its types” needs a rationale, why disabilities are explained further, but the other reasons for accessibility not.
    • L210-212: Example for an undirected writing: Why is this sentence in a section about serious games and accessibility? Same question for L213-214
  • Section 3
    • L223-226: These sentences are superfluous: The reader should be familiar with the term “systematic literature review”. If you describe a study, you do not define what a study is but you describe the methodology of the study. If you describe everything from scratch, the paper becomes too long.
    • L233: “traditional reviews“: What are traditional reviews? There are different kind of reviews, e.g. Booth and Grant (2009) identify 14 types of reviews. And BTW: Why is this important here? Does this influence your decision on the research methodology used?
    • L263: “ What methods are applied in the design of serious games?” Imprecise: methods for what? Where is the connection to accessibility?
    • L263: The appearance of research questions here is confusing. What is the connection to the contributions in L91-95? At different locations of the article the goals are explained in different ways. This needs to be aligned.
    • L265-266: “An *excellent* way to define the query string is to structure them in terms of PICO”: “excellent” is not objective but more subjective. Everybody might define “excellent” in another way. You could write “commonly used” etc.
    • L273: “four most used digital libraries” -> this statement requires a reference which proves that this are indeed the most used libraries. Or just omit “most used”.
    • Table 2: Why are there different search strings for every database?
    • L283: Table 3: This can be expressed shorter: primary study, English language, evaluation, publication between 2000-2020
    • L340: Here is a linking sentence missing: Why are you presenting the publications found per year? What is the added value for the reader? For sure, there is an added value, but it needs to be explained.
    • L350: See L340
    • 6: The arrows could be aligned in a better, prettier looking way.
  • Section 4
    • L412: Figure 8 and Table 6 are an example of doubled information. This might be shortened, e.g. by providing a column in the table indicating the percentage. This hint applies to other Table / Figure pairs as well.
    • L419: I do not understand what each category stands for. So, here it might make sense to extend the description and add for each category an example being included in the category.
    • L528: “As can be appreciated in Figure 12, the most valuable studies are those” -> wording: appreciate / valuable: for what?
    • L555 – 568: I would not write about “many” limitations if I would find only three. These are only “some” limitations, not unnecessarily weaken your work. Further, I would not enumerate limitations general to the methodology applied if there is no concrete reason to do so. IMHO, this section here could be omitted
  • Section 5:
    • L583: “Also, our exploration shows that there are open research issues in the application of accessibility guidelines by serious game designers” Have these issues been named? IMHO this would be an added value of the article.

Formal remarks

L34: The need to learn -> They need to learn? To what subject refers “They”? Or “ensure” -> I do not understand the sentence.

L 128 – 132: Please check structure of the sentence, consistency of the presentation and language (This is IMHO an example where proofreading would be beneficial): “For software, accessibility includes 1) accessing the Internet using a desktop browser or a voice browser; 2) for a platform, it includes the use of desktop, mobile phone or personal digital assistant (PDA); 3) for the environment, it includes adaptations for users working in noisy or poorly lit environments; and 4) for user capacity, it includes adaptations for users with sensory, cognitive and motor disabilities”.

L 153: “y” needs to be translated.

L263: “RQ3.What”: Missing blank

L363: If six categories are named, why not numbering these categories as it has been done in Fig. 6?

L378: The single RQs are worth of an own section with heading IMHO as it would improve readability.

L382-386: I would use bullet items as it would improve readability.

Fig. 12: Is it really necessary to have to abbreviations for each study? The letter-digit code and the common reference with name and year? I suggest limiting to one all over the paper for simplification.

L552 instead of using the word foolproof, I suggest writing about limitations of the study.

L558: “we plan to survey to evaluate” sentence structure?

Author Response

The article describes a structured literature review on accessibility in serious games. 4 frequently used literature databases are searched, at the end there are 50 articles left, which are classified more detailed. The classification is used to answer 5 research questions.

The article is focused on a research topic worthy of investigation and comes to relevant results. Before publication, however, some issues should be addressed:

We would like to express our sincere appreciation for the review of our work. Your notes have allowed us to improve the manuscript significantly and reflect on future research.

- The article is very long, because besides the actual research questions many general remarks about SLR are made, which however do not advance the actual research questions. Here a focus should be made and all explanations that are not directly necessary to answer the research questions should be removed from the article. For example, there is no need to explain the advantages and disadvantages of SLR in general terms, unless they are directly related to the research question.

Thank you very much for helping us to improve our document.  You are indeed right; we have removed sections 2.1. Accessibility, and 2.2.

- Basically the article should be checked once more for a logical structure. Some explanations are doubled, for example in section 1 and section 2, some explanations do not fit into the respective chapter. As a rule, all facts should be given only once. This also includes the information given in tables and graphics, some of which are redundant. Details about these two remarks can also be found in the further remarks.

Thank you for your suggestion. We have reviewed the entire document, eliminated repetitive information, and updated it.

- The reasons for the selection of research questions should be discussed.

Thank you very much for your comment; we have discussed the research questions and restructured them. We also added five additional questions to validate the quality of the selected articles. Furthermore, the research questions were raised because, currently, serious games have been incorporated into the teaching-learning processes; due to the increase in their use, the need arises to guarantee their accessibility to include people with disabilities in educational environments fully.

- The results found, and their implications should be better discussed. Which effects have which results? What other questions arise from these findings? What do these results imply for my own development of a serious game?

Thank you very much for your valuable feedback. To better discuss and address the results, we have separated them into two sections: results and discussion.

- Sometimes the results are only described with descriptive statistics. But it would be good to pick out examples to give the reader an impression of the typical state of accessibility in serious games. So in the end, all that remains is the impression that there are two standards, but otherwise little is done and if so, own solutions or special hardware is used.  This impression needs to be extended.

Thank you for your concern. We have updated the manuscript accordingly.

- The article should be optimized for better readability. This includes partly joingen sections (a section has more than one sentence), the use of bullet points, the addition of connection sentences between sections or subsections, the use of bold print etc.

Thank you very much for your comments; we have followed your suggestions and updated the manuscript. We included the partial linking of sections, applied the use of bullets, and added connecting sentences between sections and subsections and the use of bold type for better readability.

General remarks

The abstract does not provide a definition for accessibility. It starts with a definition for serious games and describes thereafter the methodology used for the literature review. It fails to motivate pursuing the literature research.

Thank you very much for your recommendation. We have improved our abstract.

Abstract: Nowadays, serious games, called training or learning games, have been incorporated into the teaching-learning processes; due to the increase of their use, the need to guarantee their accessibility arises to include people with disabilities in the educational environments in an integral way. There are literature reviews on video games but not on web-based serious games; are differ from the previous ones because web-based serious games are designed for different from pure fun. This literature review was conducted using the recommendations of systematic reviews proposed by Kitchenham and Petersen. Three independent reviewers searched the ACM Digital Library, IEEE Xplore, Scopus, and Web of Science databases for the most relevant articles published between 2000 and 2020. Review selection and extraction were made using an interactive team approach. We applied the study selection process's flowchart adapted from the PRISMA statement to filter in three stages. This systematic literature review provides researchers and practitioners with the current state of web-based serious games and accessibility, considering cognitive, motor, and sensory disabilities.

What are the rationales for the research questions? Why this five and not another six research questions? It would be beneficial to discuss this a bit.

Section 1:

The definition of serious games in the abstract differs from the definition in the main text. Please be consistent.

A bit lengthy. For example, the sense and purpose of a literature review does not need to be explained (L71).

Thank you very much for allowing us to improve our article.

We have updated the document with the suggested changes.

This study's first objective is to present information about the most relevant research on published web-based serious games and accessibility. This SLR contains a series of articles from the digital libraries and details the authors, the year of publication, and the Scimago Journal Rank (SJR) impact factor. The second objective is to detect the different approaches to serious web-based games for cognitive, motor, and sensory disabilities. The third objective is to identify the WCAG-based accessibility guidelines applied to serious gaming to determine trends and gaps in serious game development.

The research questions were raised because, currently, serious games have been extensively incorporated into the teaching-learning processes [24], [25]; due to the increase in their use, the need arises to guarantee their accessibility to people with disabilities in educational environments fully.

Our study examines the results of existing primary studies published on accessibility and serious games to identify current trends and open issues in the domain: our research questions and each question's purpose.

102 “to verify if any gaps could be filled with the SLR proposed in this paper.” An SLR is proposed? It is done!? Further, this seems to be part of the rational of the entire study – so, why is this not presented earlier in the paper?

Section 2

Section „2.2. Disability and its types” needs a rationale, why disabilities are explained further, but the other reasons for accessibility not.

Thank you very much! We have removed section 2.2

L210-212: Example for an undirected writing: Why is this sentence in a section about serious games and accessibility?

Thanks for the observation; we have corrected what we suggested.

Same question for L213-214

Section 3

L223-226: These sentences are superfluous: The reader should be familiar with the term “systematic literature review”. If you describe a study, you do not define what a study is but you describe the methodology of the study. If you describe everything from scratch, the paper becomes too long.

Thank you very much for your comments that have enriched our document. We have eliminated the generalities, and we relate it directly to the method used in this research.

L233: “traditional reviews“: What are traditional reviews? There are different kind of reviews, e.g. Booth and Grant (2009) identify 14 types of reviews. And BTW: Why is this important here? Does this influence your decision on the research methodology used?

Thanks for the suggestion; we have removed all explanations about "SLR."

L263: “ What methods are applied in the design of serious games?” Imprecise: methods for what? Where is the connection to accessibility?

L263: The appearance of research questions here is confusing. What is the connection to the contributions in L91-95? At different locations of the article the goals are explained in different ways. This needs to be aligned.

Thanks for the recommendation; we have applied what we suggested to align with this literature review. We have updated the entire document considering the suggestions.

L265-266: “An *excellent* way to define the query string is to structure them in terms of PICO”: “excellent” is not objective but more subjective. Everybody might define “excellent” in another way. You could write “commonly used” etc.

L273: “four most used digital libraries” -> this statement requires a reference which proves that this are indeed the most used libraries. Or just omit “most used”.

Table 2: Why are there different search strings for every database?

Thanks for the suggestion, the search chain applied is the same to achieve the most significant number of documents, but each database has its specific syntax. They are equivalent strings that seek to locate the same articles.

L283: Table 3: This can be expressed shorter: primary study, English language, evaluation, publication between 2000-2020

Thanks for the recommendation; we have removed Table 3 and placed it in the document with the suggested changes.

Inclusion criteria: The primary study must be related to 1) the type of publication in journals, conferences, books, and book chapters on accessibility in web-based serious games published from 2000 to 2020; 2) primary peer-reviewed studies, 3) written in the English language.

Exclusion criteria: The primary study 1) summarizes a keynote, a workshop introduction, or only an abstract; 2) duplicate articles from the same research from different sources.

L340: Here is a linking sentence missing: Why are you presenting the publications found per year? What is the added value for the reader? For sure, there is an added value, but it needs to be explained.

L350: See L340

Thank you again for your valuable contribution. We have improved the document. Your suggestion was placed in the Results section.

  1. Results

 In this section, we answer the research questions by:

 1) A bibliometric analysis to collect information about the authors and publication data of research growth over time, journals, conferences, books, and book chapters published on serious games and accessibility.

2) A literature review to map the studies according to serious games' concepts and the five research questions.

Bibliometric analysis

This analysis aims to respond to RQ1; in Figure 3, we summarize the inclusion stage results and highlight some findings: Figure 3 shows the evolution of scientific production, presenting the number of documents each year. The years of most scientific output in accessibility in the serious games are 2018 and 2019. We found eight papers for 2019, which corresponds to 17%, seven documents for 2018, which corresponds to 14.9%. In 2017, we found six articles, corresponding to 12.8%. In 2020, we found five documents corresponds to 10.6%; it is expected that this number tends to increase because it was done until July 2020; in 2006, 2013, and 2014 there were three documents each year that add up to 19.1%. In 2008, 2015, and 2016 there were two items each year that add up to 12.8%. Finally, in 2003, 2007, 2009, 2010, 2011, and 2012, one document added up to 12.8% each year. The annual growth rate of the published articles follows the polynomial equation (2).

y = 0.0216x2 - 0.1639x + 0.6421    (2)

Figure 3. Documents published from 2000 to 2020.

Figure 4 presents 35 conference studies representing 74.5% of the total and 12 journal articles representing 25.5%. In this review of the literature, the most significant number of studies found are concentrated in conferences. The largest number of documents are indexed in Scopus.

Figure 4. Documents by type.

The arrows could be aligned in a better, prettier looking way.

Thanks for the suggestion; we removed the image and detailed it textually.

3) we presented the classification in five aspects that were defined as follows: 1) The disability accessibility guidelines which aim to provide a standard of easy access to serious games that meets people's needs. 2) The applied solutions that involve the methods used to make serious games accessible. 3) The disability accessibility guidelines that provide input for serious play. 4) The types of disability contributions [5] include the formal study, method, system, or experience. And 5) The type of research provides for validation, solution, evaluation, feedback, and experience [5].

Section 4

L412: Figure 8 and Table 6 are an example of doubled information. This might be shortened, e.g. by providing a column in the table indicating the percentage. This hint applies to other Table / Figure pairs as well.

We appreciate the significant input in this comment; we have eliminated the duplicate information. 

L419: I do not understand what each category stands for. So, here it might make sense to extend the description and add for each category an example being included in the category.

L528: “As can be appreciated in Figure 12, the most valuable studies are those” -> wording: appreciate / valuable: for what?

L555 – 568: I would not write about “many” limitations if I would find only three. These are only “some” limitations, not unnecessarily weaken your work. Further, I would not enumerate limitations general to the methodology applied if there is no concrete reason to do so. IMHO, this section here could be omitted

Thank you very much for your comments; we have applied what we suggested.

Section 5:

L583: “Also, our exploration shows that there are open research issues in the application of accessibility guidelines by serious game designers” Have these issues been named? IMHO this would be an added value of the article.

Thank you very much for your recommendation, we have implemented what was suggested, and the document has been updated.

 Formal remarks

L34: The need to learn -> They need to learn? To what subject refers “They”? Or “ensure” -> I do not understand the sentence.

L 128 – 132: Please check structure of the sentence, consistency of the presentation and language (This is IMHO an example where proofreading would be beneficial): “For software, accessibility includes 1) accessing the Internet using a desktop browser or a voice browser; 2) for a platform, it includes the use of desktop, mobile phone or personal digital assistant (PDA); 3) for the environment, it includes adaptations for users working in noisy or poorly lit environments; and 4) for user capacity, it includes adaptations for users with sensory, cognitive and motor disabilities”.

L 153: “y” needs to be translated.

L263: “RQ3.What”: Missing blank

L363: If six categories are named, why not numbering these categories as it has been done in Fig. 6?

L378: The single RQs are worth of an own section with heading IMHO as it would improve readability. 

Dear reviewer, thank you very much. We have applied all the suggestions. The document has been updated.

L382-386: I would use bullet items as it would improve readability. 

Fig. 12: Is it really necessary to have to abbreviations for each study? The letter-digit code and the common reference with name and year? I suggest limiting to one all over the paper for simplification. 

L552 instead of using the word foolproof, I suggest writing about limitations of the study.

L558: “we plan to survey to evaluate” sentence structure?

Thank you very much. We have applied all the suggestions. The document has been updated.

Reviewer 3 Report

The accessibility of serious games, apparently only web-based serious games, has been the aim of this study. The authors presented a systematic literature review to examine the accessibility of serious games, based on a question: What accessibility evaluation standards have researchers used to evaluate serious games, and how were these standards used? The intention was to outline the issues that are particularly relevant to studies on accessibility and serious games; the way accessibility is applied in serious games, the guidelines applied, the devices used to achieve accessibility in serious games according to disability. The authors asserted that their research provides researchers and practitioners with the current state of serious games and accessibility research to help them implement and develop more inclusive serious games.

I first appreciate the effort of the authors tackling such a problem. However, the current manuscript failed to address the problem adequately. Title and Abstract are misleading. While the impression is serious games in general, but in fact only a fraction of them, web-based serious games were of the focus. The paper is not organized well. The Introduction is more puzzling than easing out the read. Filled with irrelevant information and without laying adequate background, out of sudden web content accessibility comes to picture. It is puzzling because the reader does not know that only web-based serious games are going to be studied. The next section, Background and motivation, is almost the same, verbose and filled with irrelevant info. One full page to define web accessibility is too much where the content actually does not help to tell the story.

As mentioned before, only a fraction of serious games is web-based games. In general, serious games are meant for people with some sort of disabilities. Therefore, a serious game targeting a specific disability requires careful design and implementation. The web has numerous limitations and is not a suitable platform for implementing games in general and serious games in particular. Hence, a web-based game naturally lacks several functionalities that can be easily addressed in a computer-based game.

Author Response

The accessibility of serious games, apparently only web-based serious games, has been the aim of this study. The authors presented a systematic literature review to examine the accessibility of serious games, based on a question: What accessibility evaluation standards have researchers used to evaluate serious games, and how were these standards used? The intention was to outline the issues that are particularly relevant to studies on accessibility and serious games; the way accessibility is applied in serious games, the guidelines applied, the devices used to achieve accessibility in serious games according to disability. The authors asserted that their research provides researchers and practitioners with the current state of serious games and accessibility research to help them implement and develop more inclusive serious games.

I first appreciate the effort of the authors tackling such a problem. However, the current manuscript failed to address the problem adequately. Title and Abstract are misleading. While the impression is serious games in general, but in fact only a fraction of them, web-based serious games were of the focus.

Thank you very much for your comment; we have discussed the research questions and restructured them. We also added five additional questions to validate the quality of the selected articles. Thank you so much for your constructive feedbacks and review. We have followed your suggestion about the paper and updated the manuscript.

We have updated the title to prevent it from sounding misleading.

Web-based Serious Games and Accessibility: A Systematic Literature Review

The paper is not organized well. The Introduction is more puzzling than easing out the read. Filled with irrelevant information and without laying adequate background, out of sudden web content accessibility comes to picture. It is puzzling because the reader does not know that only web-based serious games are going to be studied.

Thank you for your suggestions. We have organized the document and the paragraphs and sentences related to others to improve their reading.

The next section, Background and motivation, is almost the same, verbose and filled with irrelevant info. One full page to define web accessibility is too much where the content actually does not help to tell the story.

Dear reviewer, thank you again for your suggestions that have allowed us to improve our work. We have improved the Background and motivation, eliminating unnecessary information. We have focused on the WCAG 2.1 web accessibility guidelines because they are the most advanced and accepted mechanism for creating accessible content that is not limited exclusively to web content.

As mentioned before, only a fraction of serious games is web-based games. In general, serious games are meant for people with some sort of disabilities.

Therefore, a serious game targeting a specific disability requires careful design and implementation. The web has numerous limitations and is not a suitable platform for implementing games in general and serious games in particular. Hence, a web-based game naturally lacks several functionalities that can be easily addressed in a computer-based game. In fact, the web has numerous limitations, but if the design is considered accessible to all people, including people with some type of disability, the application will be more accessible and inclusive for a greater number of users.

Many thanks for your comments. We have focused on the WCAG 2.1 web accessibility guidelines because they are the most advanced and accepted mechanism for creating accessible content that is not limited exclusively to web content.

Reviewer 4 Report

The paper presents a systematic literature review about accessibility in Serious games. The topic seems to me be original, since the most part of game designers underestimate this issue. But in the current society it is important and I feel that the need for accessible games is growing. Title, abstract and conclusions are appropriate. The reference are cited in the text. 

The authors found a systematic review published in 2020, I suggest to give more detail about the added value of their work in comparison with the cited one. I was expected deeply discussion about the results of the review focussing not only on the number (quantitative: how many papers answer each RQ) but also on the quality of research papers (qualitative: how the papers answer to the accessibility in serious games). It would be interesting read the authors point of view. 

Moreover, the authors stated that the systematic review should help designers to implement and develop more inclusive serious games. How do they think that their work can contribute to this aim? It would be interesting read the authors' proposals.

Typos: line 259 "Research aim questions and objectives" should be a subsection. Line 386 in the list only "sensory" is in italic style, probably all item (Cognitive and Motor) should be? 

Author Response

The paper presents a systematic literature review about accessibility in Serious games. The topic seems to me be original, since the most part of game designers underestimate this issue. But in the current society it is important and I feel that the need for accessible games is growing. Title, abstract and conclusions are appropriate. The reference are cited in the text.

Dear reviewer, thank you very much for appreciating the work done and for your constructive comments. We have followed your suggestions and updated the manuscript.

The authors found a systematic review published in 2020, I suggest to give more detail about the added value of their work in comparison with the cited one.

Thank you very much. We have applied all the suggestions.

There are reviews of the literature on video games but not on web-based serious games; serious games are different from the previous ones because their educational processes allow reinforcing learning. This study identified video game SRL publications but not on serious games, so we justify this study's need to 1) Identify information on the most relevant research on web-based serious games and accessibility. 2) Identify accessibility guidelines that apply to web-based serious games. 3) To detect the different approaches to web-based serious games for cognitive, motor, and sensory disabilities. 4) To identify the WCAG-based accessibility guidelines applied to serious games to determine trends and gaps in serious games development. 5) To identify authors conducting accessibility studies on serious games.

I was expected deeply discussion about the results of the review focussing not only on the number (quantitative: how many papers answer each RQ) but also on the quality of research papers (qualitative: how the papers answer to the accessibility in serious games). It would be interesting read the authors point of view.

Thank you very much for helping us to improve our document.  We have separated the results and the discussion. Besides, to validate the quality of the selected papers, we applied a "quality assessment."

We evaluate the quality of the research articles that respond to accessibility in serious games; we apply a “quality assessment” of the selected articles. The purpose of this quality assessment (QA) is to weight the importance of each of the papers chosen when the results are discussed and to guide the interpretation of findings [6]. Each QA obtains a score of one for the fulfillment of each clause 1) Is web-based serious games accessibility detailed in the paper? 2) Is the method of evaluating the accessibility of web-based games specified in the article? 3) Does the article discuss the accessibility assessment results in web-based serious games? 4) Are the accessibility issues of the web-based serious games described? 5) Is the journal where the paper was published indexed in SCImago Journal Rank (SJR)? Table 2 presents a checklist of quality evaluation.

Table 2. Article quality evaluation checklist.

Quality Assessment questions

Answer

QA1

Is serious games accessibility detailed in the paper?

(+1) Yes/(+0) No

QA2

Is the serious games accessibility evaluation method specified in the paper?

(+1) Yes/(+0) No

QA3

Does the paper discuss any findings of serious games accessibility evaluation?

(+1) Yes/(+0) No

QA4

Are standard serious games accessibility errors described in the results?

(+1) Yes/(+0) No

QA5

Is the journal or the conference where the paper was published indexed in SJR?

(+1) if it is ranked Q1, (+0.75) if it is ranked Q2, (+0.50) if it is ranked Q3, (+0.25) if it is ranked Q4, (+0.0) if it is not indexed.

Moreover, the authors stated that the systematic review should help designers to implement and develop more inclusive serious games. How do they think that their work can contribute to this aim? It would be interesting read the authors' proposals.

This study can help software developers and designers to apply accessibility standards to have more inclusive applications even in the educational context where web-based serious games have been involved in different areas to improve and strengthen educational processes.

 Therefore we have restructured our summary tables considering the types of cognitive, motor, and sensory disabilities. The learning resources, such as a serious game, can reach many users regardless of their ability and the technologies used to access the serious game. Therefore, we detail the studies found in each variable of the study considering the types of disability so that the authors can summarize the articles and identify the strengths and weaknesses that are currently applied in serious gaming.

Typos: line 259 "Research aim questions and objectives" should be a subsection. Line 386 in the list only "sensory" is in italic style, probably all item (Cognitive and Motor) should be?

Thank you very much for the suggestion; we have applied it in our document.

Reviewer 5 Report

This paper presents a systematic review of the accessibility of serious games. The topic addressed in this paper is relevant and the review is well organized in general, but there are some concerns about certain aspects of selection criteria.

The major criticism is the inclusion criteria used in phase 4. It seems that secondary studies have been included in table 4, for instance: HL13 [REF#80], YF11 [REF#82], MZ16 [REF#72]. In the reviewer’s opinion, surveys or review articles can be considered secondary studies despite they are not systematic literature reviews. Therefore, they must be included in the introduction section (or one similar) instead of in Table 4. Please, justify the inclusion of these studies.

Additionally, other studies that already propose guidelines (e.g.: PK13, GS09, OM06, MM06) for the design of accessible serious games were also included in Table 4. It is not clear if the analysis of these papers may affect the findings of this article. In the reviewer’s opinion, the research articles (development) must be analysed separately from those papers that already suggest design guidelines. In the discussion section, the authors can compare if design recommendations are similar or not to the ones used in research papers. Please, clarify the inclusion of these studies.

As the authors expose in the limitation section 4.2., the inclusion criteria applied in this paper may put apart relevant studies in this field. In this way, most of the studies are focused on defining guidelines for increasing accessibility in serious games. However, one aspect that could be relevant when designing a serious game is what not to do. This anti-rule concept was addressed, for example, in the Urbanek et al. study. The reviewer suggests mentioning in the discussion section the possibilities of defining anti-rules to increase the accessibility of serious games.

Urbanek, P. Fikar and F. Güldenpfennig, "About the sound of bananas — Anti rules for audio game design," 2018 IEEE 6th International Conference on Serious Games and Applications for Health (SeGAH), Vienna, 2018, pp. 1-7, doi: 10.1109/SeGAH.2018.8401361.

Finally, in the Abstract section, it is not clearly stated the need for this systematic review and the final goal. After the first sentence, the authors can expose the need and aim of this review.

Author Response

This paper presents a systematic review of the accessibility of serious games. The topic addressed in this paper is relevant, and the review is well organized in general, but there are some concerns about certain aspects of selection criteria.

Dear reviewer, thank you very much for your comments. It motivates us to continue writing to receive very good ideas and contributions that have enriched our document's structure and content.

The major criticism is the inclusion criteria used in phase 4. It seems that secondary studies have been included in table 4, for instance: HL13 [REF#80], YF11 [REF#82], MZ16 [REF#72]. In the reviewer’s opinion, surveys or review articles can be considered secondary studies despite they are not systematic literature reviews. Therefore, they must be included in the introduction section (or one similar) instead of in Table 4. Please, justify the inclusion of these studies.

Thank you very much for helping us to detect the error. Thank you very much for your observation; it was an involuntary mistake. Therefore, our review was corrected and consisted of 47 documents.

Additionally, other studies that already propose guidelines (e.g.: PK13, GS09, OM06, MM06) for the design of accessible serious games were also included in Table 4. It is not clear if the analysis of these papers may affect the findings of this article. In the reviewer’s opinion, the research articles (development) must be analysed separately from those papers that already suggest design guidelines. In the discussion section, the authors can compare if design recommendations are similar or not to the ones used in research papers. Please, clarify the inclusion of these studies.

Dear reviewer, thank you very much for sharing your idea. We have applied it in section 4.2.2.

PK13 is included in WCAG 2.0. GS09, GA-SIG Guidelines to create something similar to the W3C/WAI WCAG. OM06, Guidelines from MediaLT, the rules and hints form GA-SIG and our own ideas. MM06 change Id by OA06, Section 508

As the authors expose in the limitation section 4.2., the inclusion criteria applied in this paper may put apart relevant studies in this field. In this way, most of the studies are focused on defining guidelines for increasing accessibility in serious games. However, one aspect that could be relevant when designing a serious game is what not to do. This anti-rule concept was addressed, for example, in the Urbanek et al. study. The reviewer suggests mentioning in the discussion section the possibilities of defining anti-rules to increase the accessibility of serious games.

Urbanek, P. Fikar and F. Güldenpfennig, "About the sound of bananas — Anti rules for audio game design," 2018 IEEE 6th International Conference on Serious Games and Applications for Health (SeGAH), Vienna, 2018, pp. 1-7, doi: 10.1109/SeGAH.2018.8401361.

Dear Reviewer, thank you for your suggestions. “Anti patterns”, it's a great idea, and we leave it as future work.

Finally, in the Abstract section, it is not clearly stated the need for this systematic review and the final goal. After the first sentence, the authors can expose the need and aim of this review.

Thank you very much for your feedback; we have applied what we suggested. The abstract has been improved. We have addressed it in the motivation section.

Abstract: Nowadays, serious games, called training or learning games, have been incorporated into the teaching-learning processes; due to the increase of their use, the need to guarantee their accessibility arises to include people with disabilities in the educational environments in an integral way. There are reviews of the literature on video games but not on web-based serious games; serious games are different from the previous ones because their educational processes allow reinforcing learning. This literature review was conducted using the recommendations of systematic reviews proposed by Kitchenham and Petersen. Three independent reviewers searched the ACM Digital Library, IEEE Xplore, Scopus, and Web of Science databases for the most relevant articles published between 2000 and 2020. Review selection and extraction were made using an interactive team approach. We applied the study selection process's flowchart adapted from the PRISMA statement to filter in three stages. This systematic literature review provides researchers and practitioners with the current state of web-based serious games and accessibility, considering cognitive, motor, and sensory disabilities.

  1. Background and motivation

The formal description given by [11] indicates that “serious games” have an explicit and carefully thought out educational purpose and are not intended to be played primarily for fun [12]. Serious games are educational or training games [12], while video games [13] provide a cultural outlet where more players can be included and interacted to perform activities between different users.

This study identified video game SRL publications but not on serious games, so we justify this study's need to 1) Identify information on the most relevant research on web-based serious games and accessibility. 2) Identify accessibility guidelines that apply to web-based serious games. 3) To detect the different approaches to web-based serious games for cognitive, motor, and sensory disabilities. 4) To identify the WCAG-based accessibility guidelines applied to serious games to determine trends and gaps in serious games development. 5) To identify authors conducting accessibility studies on serious games.

Round 2

Reviewer 2 Report

Dear authors, thank you very much for accommodating the comments of so many reviewers. In my opinion, the paper has improved much and is of great value for the readership. I have only some minor comments.

L16: „recommendations of systematic reviews” -> recommendations for systematic reviews?

L71-74: Isolated sentences -> I suggest removing the linebreak.

L142: Table: Cell alignments need to be unified (e.g. IEEEExplore), thinner separation lines inside of the table would enhance the formatting. This applies to other tables as well, e.g. Table 5

L162: Formatting the heading “Phase 1” bold would enhance readability as it gives orientation for the eyes. There are other opportunities to structure the text by using bold inline headings as well.

L 522: Please check reference 11, IMHO, this book is not by Elsevier, but by CRC Press? https://www.routledge.com/The-Art-of-Game-Design-A-Book-of-Lenses-Third-Edition/Schell/p/book/9781138632059

Author Response

Response to Reviewers’ Comments

 The authors thank all reviewers for their valuable and constructive feedback and review. We have applied the suggestions, learned much from their comments, and updated the manuscript.

Note: In the following responses, the revised manuscript refers to the PDF document that has yellow highlighting.

REVIEWER 2

Dear authors, thank you very much for accommodating the comments of so many reviewers. In my opinion, the paper has improved much and is of great value for the readership. I have only some minor comments.

L16: „recommendations of systematic reviews” -> recommendations for systematic reviews?

Dear reviewer, thank you so much for your constructive feedback and review. We have followed your suggestions and updated the manuscript.

This literature review was conducted using the recommendations for systematic reviews proposed by Kitchenham and Petersen.

L71-74: Isolated sentences -> I suggest removing the linebreak.

Thank you very much for your valuable feedback. We removed the linebreak.

L142: Table: Cell alignments need to be unified (e.g. IEEEExplore), thinner separation lines inside of the table would enhance the formatting. This applies to other tables as well, e.g. Table 5

Thank you very much for your suggestion to improve our article. We have reviewed all the tables and applied the thinner separation lines to improve the formatting; we have updated the document.

L162: Formatting the heading “Phase 1” bold would enhance readability as it gives orientation for the eyes. There are other opportunities to structure the text by using bold inline headings as well.

Thank you again for your recommendation to format our article. We update the suggested.

Phase 1: Identification, we include the records obtained from database searches: ACM with 92 documents, IEEE Xplore with 25 articles, Scopus with 190 articles, and WOS with 169, a total of 476 articles were extracted.

Phase 2: Screening, we apply the inclusion and exclusion criteria. Of the 476 articles, 201 articles were excluded because they were duplicated in different databases, 275 articles were included. Then, in the following filter, we excluded studies written in a language other than English, review studies, abstracts, workshops, and studies on topics other than accessibility in serious games; we excluded a total of 228 studies; finally, a total of 47 reviews passed to the next phase.

L 522: Please check reference 11, IMHO, this book is not by Elsevier, but by CRC Press? https://www.routledge.com/The-Art-of-Game-Design-A-Book-of-Lenses-Third-Edition/Schell/p/book/9781138632059

Thank you very much for helping us to improve our document.  We have corrected the suggested.

Schell J (2019) The Art of Game Design: A Book of Lenses. CRC Press

Reviewer 3 Report

I appreciate efforts of the authors to address reviewers' comments. In my case, I am happy with the answers/revisions to some of my questions/comments. But the main one has been left unanswered.

As mentioned before, web-based serious games are a small fraction of serious games deisgined to address a particular disability. Why the focus of this study is on the web-based serious games? They are not that common compared to the regular computer-based serious games. The reader is not given a comparison between existing platofroms, web-based, computer-based, mobile applications, etc. The reader is not given enough reasons why web-based serious games are improtant given the limitations of the WEB. The aim of a serious game is to adequately address a particlaur disability. This is more important factor than the game being accessible through the web. This only makes sense if the reader is given some statistics that regular computer-based serious games or other platforms failed to reach their audiences/users.

Author Response

Dear Reviewer, we express our sincere appreciation for the review of our work.

I appreciate efforts of the authors to address reviewers' comments. In my case, I am happy with the answers/revisions to some of my questions/comments. But the main one has been left unanswered.

As mentioned before, web-based serious games are a small fraction of serious games deisgined to address a particular disability. Why the focus of this study is on the web-based serious games? They are not that common compared to the regular computer-based serious games. The reader is not given a comparison between existing platofroms, web-based, computer-based, mobile applications, etc. The reader is not given enough reasons why web-based serious games are improtant given the limitations of the WEB. The aim of a serious game is to adequately address a particlaur disability. This is more important factor than the game being accessible through the web. This only makes sense if the reader is given some statistics that regular computer-based serious games or other platforms failed to reach their audiences/users.

Dear reviewer, thank you very much for appreciating the work done and for your constructive comments. We have followed your suggestions and updated the manuscript. Your comments have allowed us to improve the manuscript significantly and to reflect on future research. We think that the focus has been put on "web-based serious games" because it is an area that is growing thanks to the improvement of browsers and technologies used on the Web, which have reduced the gap between desktop application and web application.

It is also an area that has received little attention, so this work's importance is something new. This article aims not only to analyze the accessibility of serious games specifically designed for 1) People with disabilities; but the accessibility of serious games in general, which must be accessible to be inclusive. 2) even if their target audience is not people with disabilities they must be accessible.

Again, thank you very much for your valuable comments; we are very grateful.

We apply the changes in the Introduction, in section 2.1, and section 6. We attach the changes made.

  1. Introduction

The Web has changed the way people communicate and relate to each other; technology has generated a continuous impact on society and individuals' behavior. The increasing [1] access to the Web and the variety of devices that allow us to interact with it have made it possible for students to choose the tools and services that best suit their needs, to personalize the learning experience.

Figure 1 shows the Google Trends [2] search related to web applications, serious games, and mobile applications made on the Web in the last five years. We found that the term serious games and web applications are intensifying from 2019.

Figure 1. Trend of web-based serious games [2].

Serious games are “games that do not have entertainment, enjoyment, or fun as their main objective” [1, p.21]. The main objectives of serious games can be, among others, education, training, human resources management, and health improvement [4]. Web-based serious games is an area that is growing thanks to the improvement of browsers and technologies used on the Web [1], which have reduced the gap between desktop and web applications.

According to Statista [5], the game-based learning market revenue worldwide between 2018 and 2024 indicates the serious games market is expected to grow from 3.5 billion U.S. dollars to 24 billion in 2024. The trend of serious web-based games has several benefits: 1) Reinforce learning in educational processes virtually and at a distance [1]. 2) Use the applications without the need to download, install, and configure. 3) Interact with the applications at any time and space. 4) Update the application automatically with the latest version. 5) Use the applications with fewer technical problems due to software or hardware conflicts with other applications.

  • Serious games and accessibility

Accessibility in serious games [17] aims to ensure that serious games can be used by the maximum number of people to access serious games. The authors suggest applying the four principles of WCAG 2.1.

Several studies [10], [18] show a lack of commitment by designers to implement accessibility. For this reason, there is a low percentage of accessible serious games. Currently, serious games have been incorporated into the teaching-learning processes; therefore, it is essential to guarantee accessibility [19] so that the largest number of people can use them.

  1. Conclusions

3) Conduct a literature review on the accessibility of serious games for mobile and computer applications for users with disabilities.

Reviewer 5 Report

The authors have addressed the reviewer's concerns, and the additional changes in the text have improved the level of the paper.

Author Response

The authors have addressed the reviewer's concerns, and the additional changes in the text have improved the level of the paper.

Dear reviewer, thank you very much for your comments. It motivates us to continue writing to receive excellent ideas and contributions that have enriched our document's structure and content.
